# RoboMamba: Efficient Vision-Language-Action Model for Robotic Reasoning and Manipulation

**Jiaming Liu**[1], **Mengzhen Liu**[1][*], **Zhenyu Wang** [1], **Pengju An**[1], **Xiaoqi Li** [1],
**Kaichen Zhou**[1], **Senqiao Yang**[1], **Renrui Zhang** [*], **Yandong Guo**[2], **Shanghang Zhang**[1,3] ✉

[1]State Key Laboratory of Multimedia Information Processing,
School of Computer Science, Peking University; [2]AI[2]Robotics;
[3]Beijing Academy of Artificial Intelligence (BAAI)
jiamingliu@stu.pku.edu.cn, 21251282@bjtu.edu.cn, shanghang@pku.edu.cn

## Abstract

A fundamental objective in robot manipulation is to enable models to comprehend visual scenes and execute actions. Although existing Vision-Language-Action (VLA) models for robots can handle a range of basic tasks, they still face challenges in two areas: (1) insufficient reasoning ability to tackle complex tasks, and (2) high computational costs for VLA model fine-tuning and inference. The recently proposed state space model (SSM) known as Mamba demonstrates promising capabilities in non-trivial sequence modeling with linear inference complexity. Inspired by this, we introduce RoboMamba, an end-to-end robotic VLA model that leverages Mamba to deliver both robotic reasoning and action capabilities, while maintaining efficient fine-tuning and inference. Specifically, we first integrate the vision encoder with Mamba, aligning visual tokens with language embedding through co-training, empowering our model with visual common sense and robotic-related reasoning. To further equip RoboMamba with SE(3) pose prediction abilities, we explore an efficient fine-tuning strategy with a simple policy head. We find that once RoboMamba possesses sufficient reasoning capability, it can acquire manipulation skills with minimal fine-tuning parameters (0.1% of the model) and time. In experiments, RoboMamba demonstrates outstanding reasoning capabilities on general and robotic evaluation benchmarks. Meanwhile, our model showcases impressive pose prediction results in both simulation and real-world experiments, achieving inference speeds 3 times faster than existing VLA models. Our project web page: https://sites.google.com/view/robomamba-web

## 1 Introduction

The scaling up of data has significantly propelled research on Large Language Models (LLMs) [1–3], showcasing notable advancements in reasoning and generalization abilities within Natural Language Processing (NLP). To comprehend multimodal information, Multimodal Large Language Models (MLLMs) [4–8] have been introduced, empowering LLMs with the capability of visual instruction-following and scene understanding. Inspired by the strong capabilities of MLLMs in general settings, recent research aims to incorporate MLLMs into robot manipulation. On the one hand, some works [9–12] enable robots to comprehend natural language and visual scenes, automatically generating task plans. On the other hand, Vision-Language-Action (VLA) models [13–15] leverage the inherent capabilities of MLLMs, empowering them with the ability to predict low-level SE(3) poses.

---

[*]Project Lead, ✉ Corresponding author.

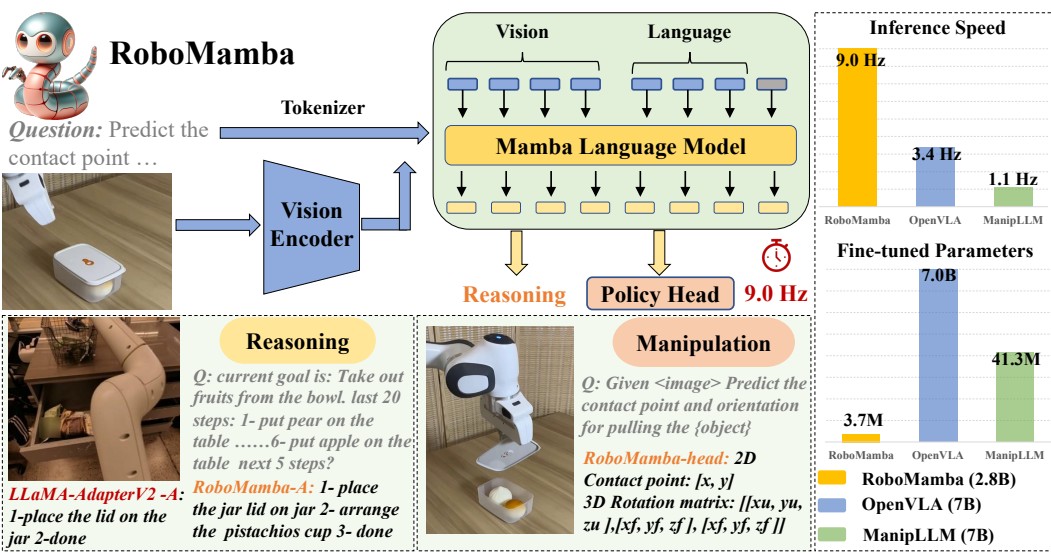

Figure 1: **Overview of RoboMamba.** RoboMamba is an efficient robotic VLA model that combines reasoning and manipulation capabilities. First, we integrate and align a vision encoder with the Mamba LLM, endowing our model with common sense and robotic-related reasoning abilities. Subsequently, we introduce an efficient fine-tuning strategy to equip RoboMamba with pose prediction abilities, requiring a few dozen minutes to fine-tune a simple policy head (3.7M parameters). In terms of inference speed, RoboMamba achieves the highest control frequency, surpassing other VLA models, running on an NVIDIA A100 GPU without any quantization or inference acceleration techniques. More real-world downstream tasks are displayed in Figure 4 and Figure 5.

Robot manipulation involves interacting with objects in dynamic environments, requiring human-like reasoning abilities to comprehend the semantic information of scenes [11, 16], alongside a robust low-level action prediction ability [17, 18]. While existing MLLM-based policies can handle a range of basic tasks, they still face challenges in two aspects. **First**, the reasoning capabilities of pre-trained MLLMs [6, 19] in robotic scenarios are found to be insufficient. As shown in Figure 1 (reasoning example), this deficiency presents challenges for fine-tuned robot MLLMs when they encounter complex reasoning tasks. **Second**, fine-tuning MLLMs and using them to generate robot manipulation actions incurs higher computational costs due to their expensive attention-based LLMs [20, 21]. To balance the reasoning ability and efficiency, several studies [22–24] have emerged in the field of NLP. Notably, Mamba [25] introduces the innovative selective State Space Model (SSM), promoting context-aware reasoning while maintaining linear complexity. Drawing inspiration from this, we raise a question: "*Can we develop an efficient robotic VLA model that possesses strong reasoning capabilities while also acquiring robot manipulation skills in a cost-effective manner?*"

To address this, we propose RoboMamba, an efficient robotic VLA that empowers the Mamba LLM to achieve robust robotic reasoning and action capabilities. As shown in Figure 1, we initially integrate a vision encoder (e.g., CLIP [26]) with Mamba to empower RoboMamba with visual common sense and robotic-related reasoning. Specifically, we proceed with alignment pre-training, activating the cross-modal connector [4, 19] to convert visual information into Mamba's token embeddings. We then unfreeze Mamba for instructions co-training, utilizing its powerful sequence modeling to comprehend high-level robotic and general instruction data. On top of this, to equip RoboMamba with SE(3) pose prediction abilities, we explore an efficient fine-tuning strategy with a simple policy head. Notably, we discover that once RoboMamba possesses sufficient reasoning capabilities, it can acquire pose prediction skills with minimal parameter fine-tuning. The fine-tuned policy head constitutes only 0.1% of the model parameters, which is 10 times smaller than existing robotic VLA approaches [15, 14]. In this way, RoboMamba can simultaneously generate robot reasoning using language responses and predict end-effector poses via the policy head.

To systematically evaluate our proposed RoboMamba, we conduct extensive experiments in both simulation and real-world scenarios. First, we assess our reasoning abilities on general and robotic

evaluation benchmarks. RoboMamba, with only 3.2B parameters, achieves competitive performance on several MLLM benchmarks and also delivers promising results on RoboVQA (42.8 BLEU-4) [27]. With its strong reasoning abilities, RoboMamba achieves state-of-the-art (SOTA) manipulation performance in the SAPIEN simulation [28], requiring only a 7MB policy head and a few dozen minutes of fine-tuning on a single A100 GPU. Moreover, RoboMamba achieves an inference speed that is 3 times faster than previous robotic VLA models [29, 15]. Additionally, we evaluate RoboMamba in real-world scenarios, where it can generate long-horizon planning and predict the end-effector pose for each atomic task. In summary, our contributions are as follows:

- We introduce RoboMamba, an efficient VLA model that integrates a vision encoder with the linear-complexity Mamba LLM, which possesses visual common sense and robotic-related reasoning abilities.

- To equip RoboMamba with action pose prediction abilities, we explore an efficient fine-tuning strategy using a simple policy head. We find that once RoboMamba achieves sufficient reasoning capabilities, it can acquire pose prediction skills with minimal cost.

- In our extensive experiments, RoboMamba excels in reasoning on general and robotic evaluation benchmarks, and showcases impressive pose prediction results in both simulation and real-world experiments.

## 2 Related work

**State Space Models (SSMs).** SSMs have become effective substitutes for transformers and CNNs due to their linear scalability with sequence length [30, 23]. Recent works [22, 31, 32] use the state space to robustly establish dependencies across long sequences. Especially, Mamba [25] designs the SSM matrices to be functions of the input, creating a learnable selection mechanism that improves adaptability and reasoning capabilities. [33–38] expand selective SSMs to vision and video tasks. Furthermore, MambaIR [39] focuses on image restoration, and PanMamba [40] addresses pan-sharpening, while DiS [41] integrates SSMs into diffusion models. These findings demonstrate that Mamba exhibits promising performance and efficiency in various visual downstream tasks. With the emergence of SSMs, we make the first attempt to introduce Mamba to address critical challenges in robotics, which demands efficient action capabilities.

**Multimodal Large Language Models.** Large language models (LLMs) have exhibited remarkable reasoning capabilities across various downstream tasks [19, 42, 2]. When addressing complex multimodal reasoning challenges, multimodal large language models (MLLMs) have shown exceptional visual understanding, i.e., BLIP-2 [43], OpenFlamingo [44], LLaMA-Adapter [19, 45], and LLaVA [46]. Additionally, the introduction of 3D MLLMs [12, 47, 48] seeks to expand the reasoning and conversational capabilities of LLMs to include the 3D modality. However, deploying LMMs is expensive due to their significant computational overhead, primarily caused by their billions of parameters. To mitigate these challenges, recent small-scale models [49, 50] demonstrate impressive performance while maintaining manageable computational costs. LLaVA-Phi [49] empowers the recently developed smaller LLM, Phi-2, for visual instruction tuning. TinyLLaVA [50] and MobileVLM V2 [51] demonstrate that high-quality training data and schemes can effectively compensate for the reasoning abilities of smaller LMMs. Furthermore, Cobra [52] innovatively utilizes an SSM-based Mamba LLM to reduce complexity and improve inference speed on common sense reasoning tasks. Different from previous works, our goal is to develop an efficient Robotic VLA model using the SSM-based language model. This model not only possesses common sense understanding but also has the capability to complete manipulation tasks effectively.

**Robot Manipulation.** Traditional robotic manipulation employs state-based reinforcement learning [53–56]. In contrast, [57, 11, 58–60] use state with visual observation as input and then make predictions. Specifically, Where2Act [61] takes visual observations and predicts on actionable pixels and movable regions in objects. Flowbot3d [57] predicts point-wise motion flow on 3D objects. Anygrasp [17] employs point cloud data to learn grasp poses on a large scale datasets. Inspired by the success of MLLMs in general scenarios [43, 44, 19, 45, 46], several VLA models [13, 16] explore utilizing their common sense reasoning capabilities to address manipulation problems. Palm-E [10] integrates multimodal encodings with LLMs, training them end-to-end for manipulation planning. VoxPoser [11] extracts affordances and constraints from MLLMs to further zero-shot predict trajectories. RoboFlamingo [14] fine-tunes MLLM on vision language manipulation dataset

to complete language-conditioned manipulation tasks. ManipLLM [15] introduces specific training scheme for manipulation tasks that equips MLLMs with the ability to predict end-effector poses. ManipVQA [62], enhancing robotic manipulation with physically grounded information processed by MLLM. In this paper, instead of fine-tuning a pre-trained MLLM, we introduce a novel efficient VLA model that possesses both robotic-related reasoning and low-level pose prediction capabilities.

## 3 RoboMamba

In Section 3.1, we introduce the preliminaries of our proposed RoboMamba, including the problem statement and a description of the language model. Subsequently, in Section 3.2 and 3.3, we describe the architecture of RoboMamba and how we empower it with common sense and robotic-related reasoning. In Section 3.4, we outline our proposed robot manipulation fine-tuning strategy, which equips our model with pose prediction skills by minimal fine-tuning parameters and time.

### 3.1 Preliminaries

**Problem statement.** For robot visual reasoning, our RoboMamba generates a language answer $L_a$ based on the image $I \in \mathbb{R}^{W \times H \times 3}$ and the language question $L_q$, denoted as $L_a = R(I, L_q)$. The reasoning answer usually contains individual sub-tasks ($L_a \rightarrow (L_a^1, L_a^2, \ldots, L_a^n)$) for one problem $L_q$. For example, when faced with a planning question like 'How to clean the table?', the response typically includes steps such as 'Step 1: Pick up the object' and 'Step 2: Place the object in the box'. For action prediction, we utilize an efficient and simple policy head $\pi$ to predict an action $a = \pi(R(I, L_q))$. Following previous works [63, 15], we use 6-DoF to express the end-effector pose of the Franka Emika Panda robot arm. The 6-DoF includes the end-effector position $a_{\text{pos}} \in \mathbb{R}^3$ representing a 3D coordinate and direction $a_{\text{dir}} \in \mathbb{R}^{3 \times 3}$ representing a rotation matrix. If training for a grasping task, we add gripper status to the pose prediction, resulting in a 7-DoF control.

**State Space Models (SSMs).** In this paper, we select Mamba [25] as our language model. Mamba consists of numerous Mamba blocks, with the most crucial component being the SSM. SSMs [21] are designed based on continuous systems, projecting the 1D input sequence $x(t) \in \mathbb{R}^L$ into a 1D output sequence $y(t) \in \mathbb{R}^L$ through a hidden state $h(t) \in \mathbb{R}^N$. An SSM consists of three key parameters: the state matrix $\mathbf{A} \in \mathbb{R}^{N \times N}$, the input matrix $\mathbf{B} \in \mathbb{R}^{N \times 1}$, and the output matrix $\mathbf{C} \in \mathbb{R}^{N \times 1}$. The SSM can be represented as follows:

$$h'(t) = \mathbf{A}h(t) + \mathbf{B}x(t); y(t) = \mathbf{C}h(t), \tag{1}$$

Recent SSMs (e.g., Mamba [25]) are constructed as discretized continuous systems using a timescale parameter $\mathbf{\Delta}$. This parameter transforms the continuous parameters $\mathbf{A}$ and $\mathbf{B}$ into their discrete counterparts $\overline{\mathbf{A}}$ and $\overline{\mathbf{B}}$. The discretization employs the zero-order hold method, defined as follows:

$$\overline{\mathbf{A}} = \exp(\mathbf{\Delta}\mathbf{A}), \tag{2}$$

$$\overline{\mathbf{B}} = (\mathbf{\Delta}\mathbf{A})^{-1}(\exp(\mathbf{\Delta}\mathbf{A}) - \mathbf{I}) \cdot \mathbf{\Delta}\mathbf{B} \tag{3}$$

$$h_t = \overline{\mathbf{A}}h_{t-1} + \overline{\mathbf{B}}x_t; y_t = \mathbf{C}h_t. \tag{4}$$

Mamba introduces the Selective Scan Mechanism (S6) to form its SSM operator in each Mamba block. The SSM parameters are updated to $\mathbf{B} \in \mathbb{R}^{B \times L \times N}$, $\mathbf{C} \in \mathbb{R}^{B \times L \times N}$, and $\mathbf{\Delta} \in \mathbb{R}^{B \times L \times D}$, achieving better content-aware reasoning. The details of the Mamba block are shown in Figure 2.

### 3.2 RoboMamba architecture

To equip RoboMamba with both visual reasoning and manipulation abilities, we start from pre-trained Large Language Models (LLMs) [25] and visual models to construct an effective VLA model architecture. As shown in Figure 2, we utilize the CLIP visual encoder [26] to extract visual features $f_v \in \mathbb{R}^{B \times N \times 1024}$ from input images $I$, where $B$ and $N$ represent batch size and tokens, respectively. In contrast to [64, 52], we do not adopt the vision encoder ensemble technique, which employs various backbones (i.e., DINOv2 [65], CLIP-ConvNeXt [66], CLIP-ViT) for image feature extraction. The ensemble introduces additional computational costs that severely impact

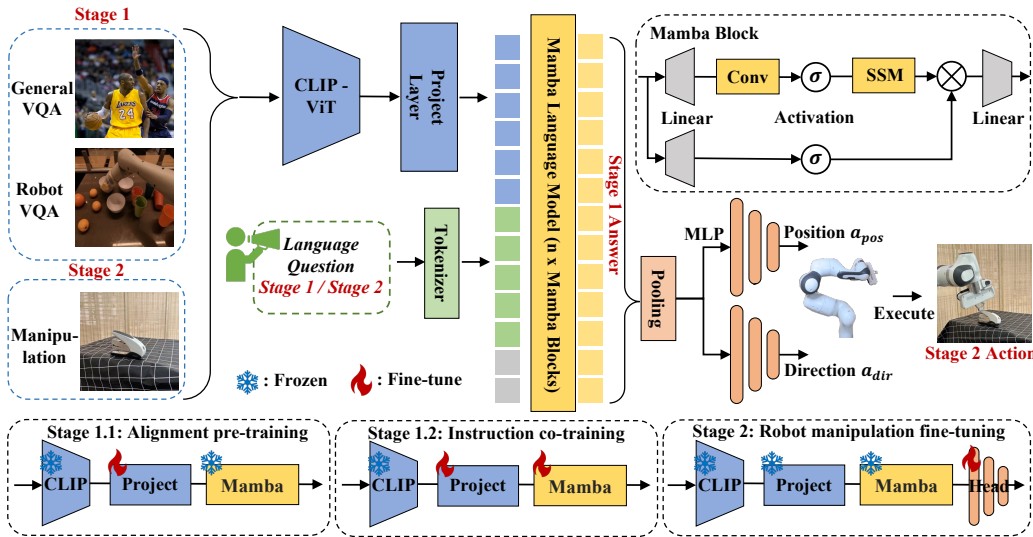

Figure 2: **Overall framework of RoboMamba.** RoboMamba projects images onto Mamba's language embedding using a vision encoder and projection layer, which is then concatenated with text tokens and fed into the Mamba model. To predict the position and rotation of the end-effector pose, we inject simple MLP policy heads and use the global token as input, which is generated through a pooling operation from the language output tokens. **Training strategy of RoboMamba.** For model training, we divide our training pipeline into two stages. In Stage 1, we introduce alignment pre-training (Stage 1.1) and instruction co-training (Stage 1.2) to equip RoboMamba with both common sense and robotic-related reasoning abilities. In Stage 2, we propose robotic manipulation fine-tuning to efficiently empower RoboMamba with low-level manipulation skills.

the practicality of VLA model in the real world. Therefore, we demonstrate that a simple and straightforward model design can also achieve strong reasoning abilities when combined with high-quality data and appropriate training strategies. To enable the LLM to understand visual features, we connect the vision encoder to the LLM using a multilayer perceptron (MLP). Through this simple cross-modal connector, RoboMamba can convert visual information into language embedding space $f_v^L \in \mathbb{R}^{B \times N \times 2560}$. Note that model efficiency is crucial in the field of robotics, as robots need to respond quickly based on human instructions. Therefore, we select Mamba as our language model due to its context-aware reasoning ability and linear computational complexity. Text prompts are encoded into embedding space $f_t \in \mathbb{R}^{B \times N \times 2560}$ using the pre-trained tokenizer, then concatenated ($cat$) with visual tokens and input into Mamba. We leverage Mamba's powerful sequence modeling to comprehend multimodal information and utilize effective training strategies to develop visual reasoning capabilities (as described in the next section). The output tokens $T_a$ are then detokenized ($det$) to produce responses in natural language $L_a$. To equip our model with both reasoning and manipulation abilities, we meticulously design a comprehensive training pipeline, which is divided into two stages. We introduce the training recipes of Stage 1 in Section 3.3 and present the robot manipulation fine-tuning in Section 3.4.

## 3.3 General and robotic-related training

After constructing the RoboMamba architecture, the next goal is to train our model to learn general vision and robotic-related reasoning abilities. As shown in Figure 2, we divide our Stage 1 training into two steps: alignment pre-training (Stage 1.1) and instruction co-training (Stage 1.2). Specifically, unlike previous MLLM training methods [19, 67, 64], we aim to enable RoboMamba to comprehend both common vision and robotic scenes. Given that the robotics field involves numerous complex and novel tasks, RoboMamba requires enhanced generalization capabilities. Therefore, we adopt a co-training strategy in Stage 1.2, combining high-level robotic data (e.g., task planning) with general instruction data. We find that co-training not only leads to more generalizable robotic policies but

also enhances the majority of general scene reasoning abilities due to the complex reasoning tasks embedded in the robotic data (demonstrated in Appendix C). The training details are shown below:

**Stage 1.1: Alignment pre-training.** We adopt LLaVA [4] filtered 558k image-text paired dataset for our cross-modal alignment. As shown in Figure 2, we freeze the parameters of the vision encoder and Mamba language model, and only update the project layer. In this way, we can align image features with the pre-trained Mamba word embedding.

**State 1.2: Instruction co-training.** In this stage, we first follow previous MLLM works [4, 64, 52] for general vision instruction data collection. We adopt the 655K LLaVA mixed instruction dataset [4], the ShareGPT4V-SFT dataset [68], or the LLaVA-Next dataset [69]. In our dataset selection, mitigating hallucination is crucial in robotic scenarios, as the robotic MLLM needs to generate task planning based on real scenes rather than imagined ones. For example, existing MLLMs might formulaically answer "open the microwave" with "step 1: find the handle," but many microwaves do not have handles. Next, we incorporate the RoboVQA dataset [27] to learn high-level robotic skills, such as long-horizon planning, success classification, discriminative and generative affordance, past description, and future prediction. During co-training, as shown in Figure 2, we freeze the parameters of the CLIP encoder and fine-tune the projection layer and Mamba on the combined instruction datasets. All outputs from the Mamba language model are supervised using the cross-entropy loss.

### 3.4 Robot manipulation fine-tuning

Building upon RoboMamba's strong reasoning ability, we introduce our robot manipulation fine-tuning strategy in this section, termed Training Stage 2 in Figure 2. Existing manipulation VLA models [29, 15, 14] require updating the projection layer and the LLM during the manipulation fine-tuning stage. While this paradigm can develop action prediction capabilities, it also breaks the inherent abilities of the pre-trained model and demands significant training resources. To address these challenges, we propose an efficient fine-tuning strategy, as shown in Figure 2. We freeze all the parameters of RoboMamba and introduce a simple policy head to model Mamba's output tokens. The policy head contains two types of MLPs that separately learn the end-effector's position $a_{\mathrm{pos}}$ and direction $a_{\mathrm{dir}}$, collectively occupying around 0.1% of the model's total parameters. Following [61], the position and direction losses are formulated as follows:

$$L_{pos} = \frac{1}{N} \sum_{i=1}^{N} |a_{\mathrm{pos}} - a_{\mathrm{pos}}^{gt}| \tag{5}$$

$$L_{dir} = \frac{1}{N} \sum_{i=1}^{N} \arccos\left( \frac{Tr\left(a_{\mathrm{dir}}^{gt\;T} a_{\mathrm{dir}}\right) - 1}{2} \right) \tag{6}$$

where $N$ represents the number of training samples, $Tr(A)$ means the trace of matrix $A$. In our open-loop simulation experiments, RoboMamba only predicts the 2D position $(x, y)$ of the contact pixel in the image, which is then translated into 3D space using depth information. We also derive the gripper's left direction (gripper z-forward) from its up and forward orientations based on geometric relationships. To evaluate this fine-tuning strategy, we generate a dataset of 10k end-effector pose predictions using the SAPIEN simulation [28]. After manipulation fine-tuning, we find that once RoboMamba possesses sufficient reasoning capabilities, it can acquire pose prediction skills with extremely efficient fine-tuning. Due to the minimal fine-tuning parameters (7MB) and efficient model design, we need only a few dozen minutes to achieve novel manipulation skills. This finding highlights the importance of reasoning abilities for learning manipulation skills and presents a new perspective: we can efficiently equip an VLA model with manipulation abilities without compromising its inherent reasoning capabilities. Finally, RoboMamba can use language responses for common sense and robotic-related reasoning, and the policy head for action pose prediction.

## 4 Experiment

In Section 4.1, we introduce our experiment settings, including dataset, implementation, and evaluation benchmark details. Subsequently, we conduct extensive experiments to demonstrate RoboMamba's reasoning and manipulation abilities in Sections 4.2 and 4.3, respectively. To thoroughly

validate the effectiveness of each method design, we perform an ablation study in Section 4.4. Finally, the qualitative results of real-world experiments are presented in Section 4.5.

## 4.1 Experiment settings

**Datasets (Stage 1)** In the alignment pre-training stage, we utilize the LLaVA-LCS 558K dataset [67], which is a curated subset of the LAION-CC-SBU dataset, supplemented with captions. During the instruction co-training stage, we combine general instruction datasets with the robotic instruction datasets. Specifically, for the general instruction dataset, we selectively adopt the LLaVA mixed instruction dataset [4], the ShareGPT4V-SFT dataset [68], or the LLaVA-Next dataset [69]. For the robotic instruction dataset, we randomly sample some image-text paired training samples from the RoboVQA [27] dataset. In our main experiments, a mixture of the LLaVA 1.5 instruction dataset and the 300K RoboVQA dataset is used during the co-training stage. Detailed descriptions of these datasets are provided in Appendix B.

**Datasets (Stage 2)** For the dataset used in the robot manipulation fine-tuning stage, we follow the data collection process of previous works [61, 15], adopting the SAPIEN engine [28] to set up an interactive simulation environment with articulated objects from PartNet-Mobility [58]. The Franka Panda Robot, equipped with a suction gripper, serves as the robotic actuator. During data collection, we randomly select a contact point **p** on the movable part and orient the end-effector's z-axis opposite to its normal vector, with a random y-axis direction to interact with the object. Successful operations are categorized as successful samples and integrated into the dataset. In the training set, we collect 10K images across 20 tasks. For evaluation, we generate 1.1K examples for the test set, comprising 20 training (seen) and 10 testing (unseen) tasks. The unseen tasks are used to evaluate the generalization capability of our model. The details of the categories are provided in Appendix B.

**Implementation details** Before training, RoboMamba loads a pre-trained CLIP/SigLIP ViT-Large [26, 70] as the visual encoder, and the 2.8/1.4B Mamba [1] model as the language model. During the alignment pre-training and instruction co-training, we conduct training for 1 epoch and 2 epochs, respectively. We utilize the AdamW optimizer with $(\beta_1, \beta_2) = (0.9, 0.999)$ and a learning rate (LR) of 4e-5. The precision of floating-point calculations is set to 16-bit. For manipulation fine-tuning, we train the model for 8 epochs, setting the LR to 1e-5 and applying a weight decay of 0.1. The floating-point precision is set to 32-bit. All experiments are conducted on NVIDIA A100 GPUs.

**Reasoning evaluation benchmarks** To evaluate reasoning capabilities, we employ several popular benchmarks, including VQAv2 [71], OKVQA [72], RoboVQA [27], GQA [73], VizWiz [74], POPE [75], MME [76], MMBench [77], and MM-Vet [78]. As detailed in Appendix E, we describe the key aspects each benchmark focuses on when assessing models in the field of robotics. Notably, we also directly evaluate RoboMamba's robotic-related reasoning abilities on the 18k validation dataset of RoboVQA, covering robotic tasks such as long-horizon planning, success classification, discriminative and generative affordance, past description, and future prediction.

**Manipulation evaluation benchmarks** To evaluate our model's manipulation capabilities, we follow previous works [57, 63, 15] and test open-loop task completion accuracy exclusively in the simulator [28]. The predicted contact point and rotation are used to interact with objects. To measure the model's performance, we use the classical manipulation success rate, defined as the ratio of successfully manipulated samples to the total test samples. A manipulation action is considered successful if the difference in the object's joint state before and after interaction exceeds a threshold of 0.1 meters. In real-world experiments, we use the Franka Panda robot to manipulate several articulated objects.

## 4.2 Reasoning quantitative results

**General reasoning.** As shown in Table 1, we compare RoboMamba with previous state-of-the-art (SOTA) MLLMs on general VQA and recent MLLM benchmarks. First, we find that RoboMamba achieves promising results across all VQA benchmarks, using only a 2.7B language model. The results demonstrate that our simple architecture design is effective. The proposed instruction co-training significantly enhance the MLLM's reasoning capabilities. For example, due to the large amount of robot data introduced during the co-training stage, our model's spatial identification performance on the GQA benchmark is improved. Meanwhile, we also test our RoboMamba on recently proposed MLLM benchmarks. Compared to previous MLLMs, we observe that our model

Table 1: Comparison of general reasoning abilities with previous MLLMs across several benchmarks. 'Res.' indicates the resolution of the input image. RoboVQA1 to RoboVQA4 represent the BLEU-1 to BLEU-4 scores, respectively. For TinyLLaVA and LLaMA-AdapterV2, we evaluate robotic reasoning abilities after fine-tuning the pre-trained MLLMs on the RoboVQA dataset.

| Method | LLM | Res. | OKVQA | VQAV2 | GQA | VizWiz | POPE | MME | MMB | MM-Vet | RoboVQA$_4$ | RoboVQA$_1$ |
|---|---|---|---|---|---|---|---|---|---|---|---|---|
| BLIP-2 [43] | 7B | 224 | 45.9 | - | 41.0 | 19.6 | 85.3 | 1293.8 | - | 22.4 | - | - |
| InstructBLIP [79] | 7B | 224 | - | - | 49.5 | 33.4 | - | - | 36 | 26.2 | - | - |
| LLaMA-AdapterV2 [45] | 7B | 336 | 49.6 | 70.7 | 45.1 | 39.8 | - | 1328.4 | - | - | 8.1 | 27.8 |
| MiniGPT-v2 [80] | 7B | 448 | 57.8 | - | 60.1 | 53.6 | - | - | - | - | - | - |
| Qwen-VL [81] | 7B | 448 | 58.6 | 79.5 | 59.3 | 35.2 | - | - | 38.2 | - | - | - |
| LLaVA1.5 [67] | 7B | 336 | - | 78.5 | 62.0 | 50.0 | 85.9 | **1510.7** | 64.3 | 30.5 | - | - |
| SPHINX [64] | 7B | 224 | 62.1 | 78.1 | 62.6 | 39.9 | 80.7 | 1476.1 | 66.9 | **36.0** | - | - |
| LLaVA-Phi [49] | 2.7B | 336 | - | 71.4 | 35.9 | - | 85.0 | 1335.1 | 59.8 | 28.9 | - | - |
| MobileVLM [82] | 2.7B | 336 | - | - | 59.0 | - | 84.9 | 1288.9 | 59.6 | - | - | - |
| TinyLLaVA [83] | 2.7B | 336 | - | 77.7 | 61.0 | - | 86.3 | 1437.3 | **68.3** | 31.7 | 29.6 | 43.5 |
| RoboMamba(Ours) | 2.7B | 224 | **63.3** | **79.6** | **64.2** | 57.1 | 86.3 | 1297.2 | 60.9 | 29.4 | **42.8** | **62.7** |
| RoboMamba(Ours) | 2.7B | 336 | 62.7 | 77.7 | 63.3 | **58.1** | **87.0** | 1335.5 | 60.7 | 31.4 | 41.8 | 61.9 |

achieves competitive results across all benchmarks. Specifically, our model achieves satisfactory results on the POPE benchmark, helping to reduce failed robot actions caused by hallucinations. Although some performances of RoboMamba are still below those of LLaVA1.5 and SPHINX, we prioritize using a smaller and faster Mamba to balance the efficiency of the robotic model. In the future, we plan to develop RoboMamba-7B for scenarios where resources are not limited.

**Robotic-related reasoning.** To comprehensively compare RoboMamba's robotic-related reasoning abilities, we benchmark it against LLaMA-AdapterV2 [45] and TinyLLaVA [83] on the RoboVQA [27] validation set. We choose LLaMA-AdapterV2 as a baseline because it serves as the base model for the current state-of-the-art (SOTA) robot MLLM, ManipLLM [15]. Meanwhile, TinyLLaVA is chosen as a representative tiny MLLM, enabling a comparison of robotic-related reasoning abilities. For a fair comparison, we load the pre-trained parameters of both LLaMA-AdapterV2 and TinyLLaVA and fine-tuned the baseline models on the RoboVQA training set for two epochs, using their official instruction-tuning method. As shown in Table 1, RoboMamba achieves superior performance across BLEU-1 to BLEU-4. The results indicate that our model possesses advanced robotic-related reasoning capabilities and confirms the effectiveness of our training strategy. In addition to higher accuracy, our model achieves inference speeds 7 times faster than LLaMA-AdapterV2 and ManipLLM, which can be attributed to the content-aware reasoning ability and efficiency of the Mamba language model [25]. Finally, we visualize the qualitative results in Figure 4.

### 4.3 Manipulation quantitative results

**Baselines.** To evaluate RoboMamba's manipulation abilities, we compare our model with four baselines: UMPNet [63], Flowbot3D [57], RoboFlamingo [14], and ManipLLM [14]. Before comparison, we reproduce all baselines and train them on our collected dataset. For UMPNet, we execute manipulation on the predicted contact point, with the orientation perpendicular to the object's surface. Flowbot3D predicts motion direction on the point cloud, selecting the largest flow magnitude as the interaction point and using the direction of the flow to represent the end-effector's orientation. RoboFlamingo and ManipLLM separately load the pre-trained parameters of OpenFlamingo [44] and LLaMA-AdapterV2 [45], and follow their respective fine-tuning and model updating strategies.

**Results.** As shown in Table 2, our RoboMamba achieves a 7.0% improvement on seen tasks and a 2.0% improvement on unseen tasks compared to the previous SOTA ManipLLM. Moreover, our method showcases SOTA performance across 14 of 20 seen tasks, highlighting its effectiveness and stability in predicting action poses. For unseen tasks, the recent three MLLM-based methods—RoboFlamingo, ManipLLM, and our method—all achieved promising performance. The results demonstrate that leveraging the strong generalization abilities of MLLMs can effectively improve the policy's generalization ability while enhancing accuracy on unseen objects. Regarding efficiency, RoboFlamingo updates 35.5% (1.8B) of the model parameters, ManipLLM updates an adapter (41.3M) comprising 0.5% of the model parameters, whereas our fine-tuned simple policy head (3.7M) only constitutes 0.1% of the model parameters. RoboMamba effectively updates 10 times fewer parameters than previous MLLM-based methods while achieving seven times faster inference speeds. The results reveal that our RoboMamba not only possesses strong reasoning abilities but also can acquires manipulation capabilities in a cost-effective manner.

Table 2: Comparison of the success rates between RoboMamba and baselines across various training (seen) and test (unseen) tasks. The representation for each task icon is shown in Table 3.

| Method | Seen Categories | | | | | | | | | | | | | | | |
|---|---|---|---|---|---|---|---|---|---|---|---|---|---|---|---|---|
| UMPNet [63] | 0.28 | 0.41 | 0.25 | 0.20 | 0.49 | 0.20 | 0.35 | 0.57 | 0.51 | 0.25 | 0.66 | 0.17 | 0.17 | 0.26 | 0.27 | 0.40 |
| FlowBot3D [57] | 0.50 | 0.53 | 0.26 | 0.36 | 0.34 | 0.36 | 0.54 | 0.26 | 0.12 | 0.34 | 0.41 | 0.23 | 0.36 | 0.30 | 0.17 | 0.37 |
| RoboFlamingo [14] | 0.48 | 0.51 | **0.50** | 0.35 | 0.11 | 0.47 | 0.54 | 0.35 | 0.19 | 0.46 | 0.18 | 0.64 | 0.26 | 0.42 | 0.15 | 0.87 |
| ManipLLM [15] | 0.68 | 0.62 | 0.45 | 0.74 | 0.42 | 0.25 | 0.61 | **0.66** | **0.56** | 0.52 | 0.50 | 0.42 | **0.64** | 0.76 | **0.63** | 0.60 |
| RoboMamba (Ours) | **0.81** | **0.73** | 0.33 | **0.85** | **0.86** | **0.60** | **0.81** | 0.42 | **0.56** | **0.54** | **0.68** | **0.81** | 0.26 | **0.86** | 0.39 | **0.91** |

| Method | Seen Categories | | | | AVG | Unseen Categories | | | | | | | | | | AVG |
|---|---|---|---|---|---|---|---|---|---|---|---|---|---|---|---|---|
| UMPNet [63] | 0.27 | 0.37 | 0.19 | 0.60 | 0.34 | 0.32 | 0.36 | 0.18 | 0.37 | 0.21 | 0.12 | 0.04 | 0.53 | 0.28 | 0.13 | 0.26 |
| FlowBot3D [57] | 0.21 | 0.57 | 0.29 | 0.45 | 0.35 | **0.36** | 0.36 | 0.18 | 0.30 | 0.21 | 0.50 | 0.13 | 0.53 | 0.28 | 0.09 | 0.30 |
| RoboFlamingo [14] | 0.20 | 0.42 | **0.58** | 0.60 | 0.41 | **0.36** | 0.62 | 0.64 | 0.33 | 0.14 | 0.34 | 0.44 | 0.66 | **0.41** | 0.31 | 0.43 |
| ManipLLM [15] | **0.41** | **0.78** | 0.41 | 0.59 | 0.56 | 0.21 | 0.25 | **0.79** | 0.76 | 0.52 | **0.76** | 0.43 | **0.85** | 0.26 | 0.52 | 0.51 |
| RoboMamba(Ours) | 0.40 | 0.55 | 0.37 | **0.80** | **0.63** | 0.19 | 0.23 | 0.67 | 0.66 | **0.57** | 0.45 | **0.65** | 0.68 | 0.30 | **0.93** | **0.53** |

Figure 3: **Ablation study a)** The impact of LLM on reasoning abilities. **Ablation study b)** The impact of reasoning ability on manipulation accuracy.

## 4.4 Ablation study

**The impact of LLM on reasoning abilities.** As shown in Figure 3 a), we explore the impact of different LLMs on general and robotic-related reasoning abilities. Given that efficiency is crucial in robotic tasks and directly affects the practicality of policy models, we compare Mamba-2.7B with other linear complexity LLMs. For all experiments, we utilize the same training data and strategy. Compared with RWKV-3B [24], Mamba-2.7B achieves significant improvements on both common sense and robotic-related reasoning benchmarks. The results demonstrate that the Mamba-2.7B model not only possesses linear complexity but also efficiently acquires strong reasoning abilities through our proposed training strategy. Meanwhile, our proposed RoboMamba VLA framework and training strategy can also be adapted to other, more advanced linear-complexity LLM models.

**The impact of reasoning abilities on manipulation accuracy.** We explore whether utilizing MLLMs with different reasoning abilities affects manipulation skill learning. For a fair comparison, we use the same manipulation fine-tuning strategy, injecting and fine-tuning a simple MLP policy head after the MLLM (while freezing other parameters). We compare our RoboMamba-2.7B (Ours-2.7B) with OpenFlamingo, LLaMA-AdapterV2, and our RoboMamba-1.4B. As shown in Figure 3 b), Ours-2.7B achieves promising results compared with other methods, which is proportional to its reasoning ability. Meanwhile, Ours-2.7B (w/o C) indicates that we did not use the instruction co-training method, omitting the robotic-related RoboVQA dataset during fine-tuning. We find that this also impacts the accuracy of manipulation, especially reducing the model's generalization ability when facing unseen objects. The results confirm our finding: fine-tuning an MLLM to learn robot skills does not require extensive resources; it only requires that the MLLM possesses strong robotic-related reasoning abilities. Additionally, we present more ablation studies in Appendix C, including explorations of different vision encoders, training datasets, and policy head design.

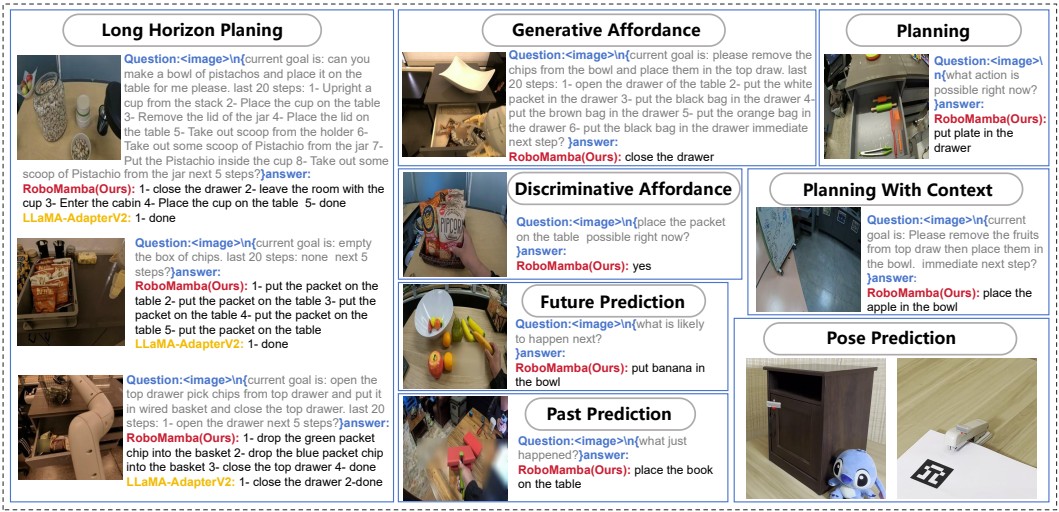

Figure 4: The visualization of RoboMamba's abilities across various robotic downstream tasks in real-world scenarios, including task planning, long-horizon planning, discriminative and generative affordance, past and future prediction, and low-level pose prediction.

## 4.5 Real-world experiments

As shown in Figure 4, we visualize RoboMamba's reasoning results across various robotic downstream tasks. For task planning, compared to LLaMA-AdapterV2, RoboMamba demonstrates more accurate and long-horizon planning abilities, thanks to its strong reasoning capabilities. For a fair comparison, we also fine-tuned the baseline LLaMA-AdapterV2 on the RoboVQA dataset. Additionally, RoboMamba accurately performs fundamental robotic tasks such as affordance generation and discrimination, proving that it can understand robotic scenes. Notably, our model also possesses past and future prediction capabilities, further highlighting its robust reasoning capabilities. Prediction of past and future actions is crucial in robotic manipulation, as it not only enables effective rethinking of past failure actions but also enhances the robustness of future manipulation pose generation. For low-level action, we use a Franka Emika robotic arm to interact with various household objects. Due to the direct visualization of the gripper causing occlusion, we project RoboMamba's predicted 6 DoF pose onto a 2D image, using a red dot to indicate the contact point and the end-effector to show the rotation, as shown in the bottom right corner of the figure.More real-world demonstrations are provided in Appendix D and the supplementary video file. Meanwhile, as shown in Figure 5, we also visualize the failure cases of RoboMamba's predictions in both reasoning and manipulation tasks.

## 5 Conclusion and future plan

In this paper, we introduce RoboMamba, an efficient VLA model that combines a vision encoder with the linear-complexity Mamba LLM, equipped with visual common sense reasoning and robotic reasoning abilities. Based on our RoboMamba, we can impart new manipulation skills to the model by fine-tuning a simple policy head (0.1% of the model) in a few dozen minutes. This finding reveals how to efficiently equip an VLA model with manipulation abilities without compromising its inherent reasoning capabilities. Finally, RoboMamba excels in reasoning on both general and robotic-related evaluation benchmarks and showcases impressive pose prediction results. Regarding limitations, while our proposed RoboMamba achieves efficient inference speed, its reliance on a 2.7B LLM leads to limitations on certain complex reasoning tasks when compared to MLLMs built on 7B/13B LLMs. Looking ahead, our future work will focus on two main directions. **1)** We plan to adapt the RoboMamba VLA framework and training strategy to more advanced linear-complexity LLM models to further enhance its reasoning and manipulation capabilities. **2)** Constructing a 4D Robot VLA model [84, 48, 85], as 3D point cloud and temporal data contain more robotics-specific information that aids in predicting robust low-level actions.

**Acknowledgements.** This work was supported by the National Natural Science Foundation of China (62476011).

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

# A Appendix

Due to space limitations, we provide additional details of the proposed method in this supplementary material. In Appendix B, we offer a more detailed description of our training dataset, including alignment pre-training, instruction co-training, and robot manipulation fine-tuning datasets. Additional ablation studies are presented in Appendix C, exploring the impact of different image encoders on reasoning ability, the impact of different training datasets on reasoning ability, and the effect of various head designs on manipulation fine-tuning. In Appendix D, we show additional qualitative results across multiple robot-related downstream tasks. Finally, we provide the metric selection rationale and the usage of prompts during testing in Appendix E.

# B Dataset description

**Stage 1.1: Alignment pre-training dataset**

**1) LLaVA-LCS 558K**: This LLaVA Visual Instruct Pretrain LCS-558K dataset is a curated subset of the LAION/CC/SBU dataset, specifically filtered to achieve a more balanced distribution of concept coverage. Additionally, it includes captions paired with BLIP synthetic captions for reference purposes.

**Stage 1.2: Instruction co-training dataset.**

**1) LLaVA-v1.5 655K**: This dataset is a mixture of ten distinct datasets, including LLaVA [4], ShareGPT [86], VQAv2 [71], GQA [73], OKVQA [72], OCRVQA [72], A-OKVQA [87], TextCaps [88], RefCOCO [89, 90], and Visual Genome (VG) [91]. This mix dataset is also one of the most renowned datasets used for instruction tuning in several works [4, 67].

**2) ShareGPT4V-SFT dataset**: The ShareGPT4V-SFT dataset is similar to LLaVA-v1.5 655K [67], except that the 23K detailed description entries in LLaVA-1.5-SFT are replaced with detailed captions randomly sampled from the 100K ShareGPT4V data [86].

**3) LLaVA-Next dataset**: Compared to LLaVA-v1.5 655K [67], LLaVA-Next [69] enhances the instruction data mixture by prioritizing high-quality user instruction data and expanding multimodal document/chart data sources. For high-quality user instruction data, LLaVA-Next ensure task diversity and response quality by using existing GPT-V data (LAION-GPT-V and ShareGPT-4V) and a carefully curated 15K visual instruction dataset from LLaVA demos. Additionally, LLaVA-Next replaces TextCaps with DocVQA and SynDog-EN to improve OCR capability.

**4) RoboVQA 800K**: In co-training, we use this dataset to enhance our model's robot-related reasoning abilities. RoboVQA [27] comprises realistic data collected by performing various user requests and using multiple embodiments, such as robots, humans, and humans with grasping tools. This dataset includes 5,246 long-horizon episodes and 92,948 medium-horizon episodes of robotic tasks, each paired with image and text prompt inputs. In our experiments, we randomly select 300K image-text paired instruction samples from RoboVQA to construct the co-training dataset.

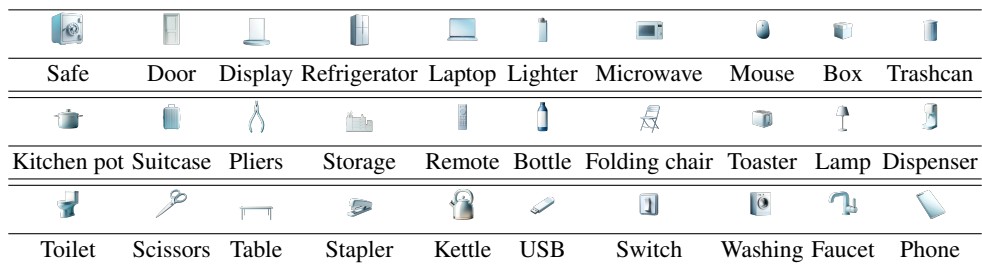

| | | | | | | | | | |
|---|---|---|---|---|---|---|---|---|---|
| Safe | Door | Display | Refrigerator | Laptop | Lighter | Microwave | Mouse | Box | Trashcan |
| Kitchen pot | Suitcase | Pliers | Storage | Remote | Bottle | Folding chair | Toaster | Lamp | Dispenser |
| Toilet | Scissors | Table | Stapler | Kettle | USB | Switch | Washing | Faucet | Phone |

Table 3: Representation of each category icon.

**Stage 2: Robot manipulation fine-tuning dataset.**

**Representation for Each Category Icon** In Table 3, we provide an overview of the meaning of each category icon presented in Table 2 of the main paper. Following Partnet [58], different tasks are designed for each category. For instance, opening the door or control panel of a refrigerator,

opening the cap of a bottle, and rotating the lid of a box. The detailed task design can be found at: https://sapien.ucsd.edu/browse.

**Simulator Data Collection** In the simulator, we use a Franka Panda Robot with a suction gripper as the robotic actuator. During data collection, we randomly select a contact point on the movable part of the object and orient the end-effector's z-axis opposite to the object's normal vector, with a random y-axis direction to interact with the object. Successful operations are categorized as successful samples and integrated into the training dataset. For the training set, we collect 10K images across 20 categories, including Safe, Door, Display, Refrigerator, Laptop, Lighter, Microwave, Mouse, Box, Trash Can, Kitchen Pot, Suitcase, Pliers, Storage Furniture, Remote, Bottle, Folding Chair, Toaster, Lamp, and Dispenser. For testing, we use a set of 1.1K images that include both seen categories from training and unseen categories, such as Toilet, Scissors, Table, Stapler, Kettle, USB, Switch, Washing Machine, Faucet, and Phone. Regarding the variation between training and testing data, we followed the data collection settings of where2act [61] and ManipLLM [15]. The specific variations can be divided into two aspects: 1) Asset Variation and 2) State Variation.

1) Asset Variation: We use 20 categories from PartNet [58] for seen objects and reserve the remaining 10 categories for unseen objects to analyze if RoboMamba can generalize to novel categories. Specifically, we further divide the seen objects into 1037 training shapes and 489 testing shapes, using only the training shapes to construct the training data. Thus, the shapes of the seen objects encountered during training and testing are different. For unseen categories, there are a total of 274 shapes, which are used exclusively in the testing data.

2) State Variation: We observe the object in the scene from an RGB-D camera with known intrinsics, mounted 4.5-5.5 units away from the object, facing its center. The camera is located at the upper hemisphere of the object with a random azimuth between [-45, 45] and a random altitude between [30, 60]. Since the tasks involve 'pulling,' we also initialize the starting pose for each articulated part randomly between its rest joint state (fully closed) and any position up to half of its joint state (half-opened). These state settings are utilized for both training and testing data, aiming to boost the model's generalization ability.

## C    Additional ablation study

Table 4: Ablation study of different image encoders on reasoning abilities.

| Encoder | Image Resolution | OKVQA | GQA | POPE | RoboVQA(BLEU-4) |
|---------|------------------|-------|-----|------|-----------------|
| CLIP | 224 x 224 | 46.7 | 50.7 | 79.7 | 35.2 |
| XCiT | 224 x 224 | 63.3 | 64.2 | 86.3 | 42.8 |
| CLIP | 336 x 336 | 62.7 | 63.3 | 87.0 | 41.8 |
| SigLIP | 384 x 384 | 62.4 | 64.4 | 86.0 | 40.6 |

**The impact of different image encoders on reasoning abilities** In this section, we replace the CLIP encoder used in our initial submission with other linear-complexity encoders, such as XCiT [92]. Additionally, we supplement our experiments by using SigLIP [70] as an image encoder. As shown in Table 4, we analyze the impact of different image encoders and input resolutions on reasoning abilities. The training dataset and strategy remain consistent with those in our main experiment. The results indicate that the choice of image encoder and input resolution does not significantly impact reasoning ability within our RoboMamba VLA framework. However, using an image encoder without cross-modality alignment (i.e., XCiT) presents challenges in converting image tokens to LLM language embeddings. Although our training process includes an alignment pre-training stage, this primarily trains the projection layer. Therefore, in future work, we aim to develop a robotics-specific image encoder capable of projecting image tokens into language embeddings while maintaining linear computational complexity to further improve inference speed.

**The impact of training datasets on reasoning abilities** As shown in Table 5, we examine the impact of different training datasets on common sense and robotic-specific reasoning abilities. Specifically, we conduct these experiments using $224 \times 224$ input images and the CLIP vision encoder. First, we observe that Ex2 outperforms Ex1 across two benchmarks, confirming that incorporating robotic instruction data can effectively enhance specific reasoning abilities. Similarly, comparing Ex3 and Ex4 shows comparable results, though performance on the GQA benchmark declines. However, in

Table 5: Ablation study of training strategies on MLLM reasoning benchmarks.

| | LLaVA 1.5 | ShareGPT4V-SFT | LLaVA-Next | Robo-300k | GQA | POPE | RoboVQA$_4$ |
|---|---|---|---|---|---|---|---|
| Ex1 | ✓ | - | - | - | 65.3 | 85.6 | 26.5 |
| Ex2 | ✓ | - | - | ✓ | 64.2 | 86.3 | 42.8 |
| Ex3 | - | ✓ | - | - | 64.3 | 85.2 | 26.7 |
| Ex4 | - | ✓ | - | ✓ | 62.1 | 85.5 | 42.5 |
| Ex5 | - | - | ✓ | - | 62.9 | 86.6 | 25.1 |
| Ex6 | - | - | ✓ | ✓ | 60.8 | 85.4 | 43.0 |

our generated descriptions within robotic scenes, we find that the inclusion of robotic instruction data enhances understanding of geometric relationships. Consequently, we plan to propose a robotic-specific geometric reasoning benchmark to more accurately assess the spatial reasoning capabilities of VLA models. Finally, comparing Ex1, Ex3, and Ex5, we find that using more advanced general instruction datasets does not yield significant performance improvements, which may be due to the model capacity of the 2.7B LLM.

**The impact of policy head designs on manipulation accuracy** As shown in Table 6, we explore the impact of different policy head designs on manipulation skill learning. In this table, MLP×1 means using only one MLP heads to predict the position and direction of the end-effector pose. MLP×2 means using one shared head to predict direction and another head to predict position separately. (SSM block+MLP)×2 is similar to MLP×2 but adds a State Space Model (SSM) block before the MLP to increase the parameter count of the policy head. The experimental results show that the manipulation accuracy across the three configurations is quite similar, indicating that the parameter count of the fine-tuning policy head has small impact on the results. Combined with Figure 3 b), this further supports our finding that once RoboMamba achieves sufficient robotic reasoning capabilities, it can acquire pose prediction skills at a low cost, regardless of the policy head design.

Table 6: Ablation study of policy head design on manipulation dataset.

| result | MLP×2 | MLP×1 | (SSM block+MLP)×2 |
|---|---|---|---|
| Acc (Seen) | 63.7% | 62.1% | 63.2% |
| Parameters | 3.7M | 1.8M | 45.2M |
| Percentage | 0.11% | 0.05% | 1.3% |

# D  Additional real-world experiments

We conduct real-world experiments involving interactions with various household objects using a Franka Emika robotic arm. We modify the finger gripper by attaching double-sided tape to convert it into a suction gripper, providing the gripper head with adhesive properties. The video demonstrations are included in the supplementary video file. As shown in Figure 5, we visualize our model's reasoning results on a series of robotic downstream tasks, including long-horizon planning, discriminative affordance, generative affordance, past description, and future prediction. Additionally, failure cases in reasoning are illustrated in Figure 6. Compared to the ground truth, RoboMamba demonstrates limitations in reasoning ability on some complex tasks, occasionally misinterpreting the current task objective or the target manipulated object.

# E  Reasoning evaluation bencharks

- **VQAv2 and OKVQA**: These benchmarks are utilized to assess the model's proficiency in basic vision question answering, which is a foundational skill in embodied AI. This ability ensures that the model can understand and respond to visual content effectively.

- **POPE and VizWiz**: These benchmarks are chosen to evaluate the model's capability to answer questions without falling prey to visual illusions or ambiguities. This aspect is crucial for avoiding significant errors in robotic applications.

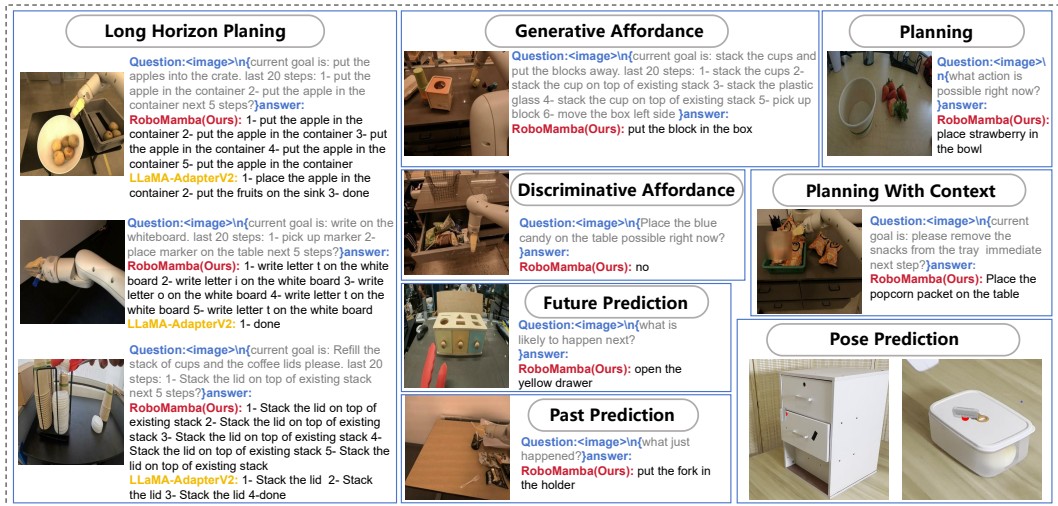

Figure 5: Additional visualization of RoboMamba's abilities across various robotic downstream tasks in real-world scenarios, including task planning, long-horizon planning, discriminative and generative affordance, past and future prediction, and low-level pose prediction.

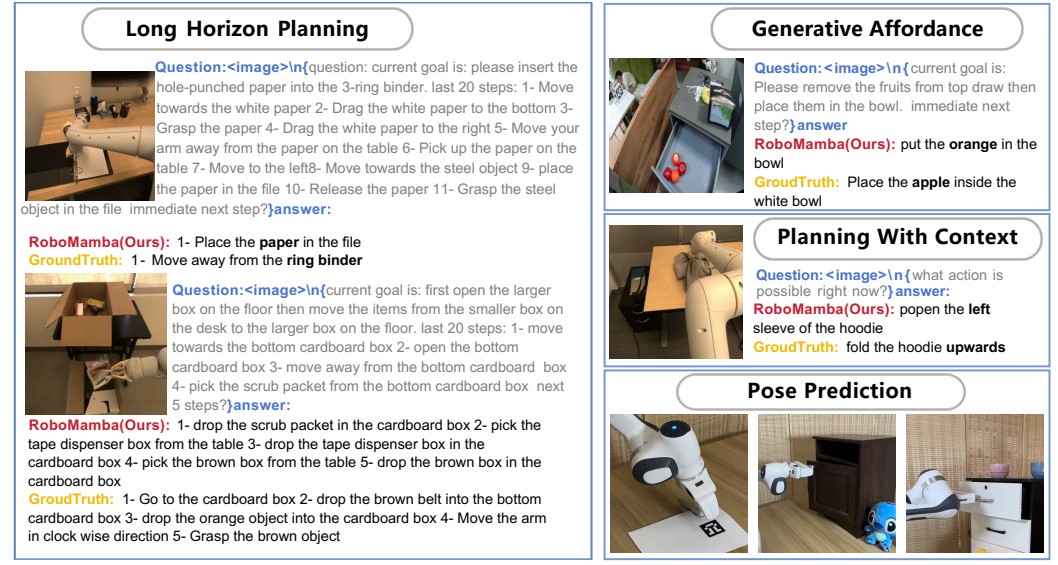

Figure 6: The visualization of reasoning failure cases. In the bottom right corner of the image, we re-select the qualitative results from our real-world demonstration. Additionally, we replace the red dot and virtual end-effector with a physical Franka Panda Robot.

- **GQA**: These benchmarks are employed to test the model's ability to identify and comprehend the types and positions of important objects within an image. Such spatial identification skills are vital for tasks related to robotic manipulation and interaction with the environment.
- **RobotVQA**: This benchmark is used to assess the model's ability to plan and understand actions based on both textual and visual inputs. This skill is indispensable in the realm of robotics, where understanding and executing complex actions is necessary.
- **MM-Vet, MME and MMB**: These benchmarks are utilized to evaluate multimodal large language models's ability to integrate on complex multi-modal tasks including Recognition, Spatial awareness, OCR, and Math. All of them contain a wealth of evaluation indicators, such as perception and cognition, which can fully demonstrate the performance of the model under different tasks, and this performance is the best embodiment of the comprehensive application performance of multimodal large language models(MLLM).

