# OpenReview forum: "RoboMamba: Efficient Vision-Language-Action Model for Robotic Reasoning and Manipulation"
_NeurIPS.cc/2024/Conference — NeurIPS 2024 poster_

### Official Review · Reviewer_16K4 · 2024-07-10

**Soundness:** 2
**Presentation:** 3
**Contribution:** 2
**Rating:** 5
**Confidence:** 4

**Summary:**

This paper introduces RoboMamba, a Multimodal LLM, for robotic reasoning and low-level manipulation using the popular State-Space Model architecture. RoboMamba introduces a multi-stage joint language and vision training pipeline with robot-specific data fine-tuning. RoboMamba demonstrates strong reasoning performance on general and robot evaluation benchmarks while using a small amount of fine-tuned parameters.

**Strengths:**

- Paper is well-written and easy to follow
- Well-motivated and comprehensive set of experiments on different general reasoning benchmarks and robot manipulation specific benchmarks
- Training dataset and evaluation details are clearly stated

**Weaknesses:**

- There is not much technical novelty in the method. It seems like RoboMamba is combining CLIP with SSMs and fine-tuning on robot-specific data (RoboVQA) which is a generic pipeline that several other MLLMs works have proposed. There are no new insights from the paper.
- The parameter-efficiency argument is mainly due to the use of SSM blocks instead of Transformer blocks. I think this is already obvious given the architecture design and is not a major contribution of this work.
- Using the term “real-world experiments” is confusing because it implies there is a physical robot executing a trained policy to perform some task which is not the case in this paper. Please provide quantitative results for the real-world reasoning experiments if possible and also provide some failure cases of your method otherwise it seems like the qualitative examples are cherry-picked.
- Please provide explanation as to why the RoboMamba training strategy may help learn better representation or capture knowledge that baselines or prior work fail to.
- Ablation study in Table 4 seem to suggest that the different fine-tuning strategies doesn’t seem to provide much benefit in terms of the general reasoning benchmarks. The improvement in downstream performance is minimal if you take out certain datasets. What about for the robot manipulation benchmarks?
- It would be nice to see experiments showing the scalability of RoboMamba, e.g. more SSM blocks, more data, etc.

**Questions:**

- Why would the baselines perform worse than RoboMamba if they're trained on the same data, unless the main factor is the architectural choice?
- Similarly, why would RoboMamba perform better than general VLMs on those benchmarks in Table 1 and how is this comparable if they are not trained on the same data?

**Limitations:**

The authors mention that using MLLMs for task planning occasionally results in hallucinations.

---

> ### Author Rebuttal · Authors · 2024-08-07
>
> - **_(Weakness 1) Technical Novelty._** Thank you for your detailed comments. We would like to reiterate the technical novelties of RoboMamba in two aspects: 1) an all-round and generalist robotic MLLM framework and 2) a robotic-specific training strategy.
>
> **1) An all-round and generalist robotic MLLM framework.** The framework designs goal of RoboMamba is to equip the model with high-level reasoning, low-level manipulation actions, and efficiency, all of which are critical attributes in the robotics field. RoboMamba is the first Robotic MLLM framework to effectively balance these three important attributes. Transferring an SSM-based LLM with a completely different architecture into a new task is not trivial. To achieve this, we innovatively chose the Mamba model, which has strong sequence modeling and linear complexity. Conversely, as shown in **Q2 2) and Table 3 of the global rebuttal response**, simply replacing the LLM in our proposed RoboMamba with another efficient LLMs results in a decline in both high-level reasoning and low-level manipulation capabilities. Meanwhile, as highlighted in **global rebuttal Q1**, RoboMamba achieves a more efficient reasoning speed than previous robotic MLLMs. The results demonstrate that building a Robotic MLLM with the aforementioned three attributes is a highly meaningful exploration and innovation, requiring appropriate architectural design and a specialized training strategy.
>
> **2）Robotic-specific training strategy.** We do not simply use a robot-specific dataset (i.e., RoboVQA and manipulation dataset); instead, we meticulously design a training strategy tailored to the robotic domain and our proposed framework. Our training strategy comprises three steps: alignment pre-training, instruction co-training, and robot manipulation fine-tuning. In the instruction co-training stage, we innovatively combine high-level general VQA data with high-level robotic data for integrated training, demonstrating that the fusion of these two data types can mutually enhance each other. As shown in Table 1 and Figure 3 a) of the submitted paper, the results demonstrate that RoboMamba possesses both visual common sense and robot-related reasoning abilities, which previous robotic MLLMs lacked. The performance improvement is due to the robotic data containing a large amount of complex multi-step reasoning data, while the general VQA data includes rich visual scene understanding data. These complement each other, providing RoboMamba with better representation. However, we did not convert low-level manipulation data to high-level data for mixed training. As shown in Table 1 below, we find that the previous SOTA method [12], which uses low-level manipulation data for fine-tuning LLM, leads to a decline in reasoning abilities compared to its pre-trained MLLM model (LLaMA-AdapterV2 [48]). In contrast, for our robot manipulation fine-tuning, we innovatively find that once RoboMamba possesses sufficient reasoning capabilities, it can acquire pose prediction skills with minimal policy head parameter fine-tuning, without compromising the inherent abilities of the MLLM.
>
> |**Table1:**|GQA|OKVQA|VQAV2
> -|-|-|-
> ManipLLM[12]|42.2|30.1|56.1
> LLaMA-AdapterV2[48]|45.1|49.6|70.7
>
> - **_(Weakness 2) Parameter-efficiency._** Sorry for the confusion. As indicated in Lines 16, 54, and 71 of the submitted paper, our use of "parameter-efficiency" refers to the minimal parameter fine-tuning (MLP Policy Head) required for learning low-level manipulation skills. It does not suggest that SSM-based LLMs are more parameter-efficient than Transformer-based LLMs.
>
> - **_(Weakness 3) Real-world experiments._** As shown in the bottom right corner of Figure 4 (Pose Prediction), we use a physical Franka Panda robotic arm to manipulate real-world objects. The demonstration video is available in our supplementary material. We apologize for the confusion. We use a red dot to indicate the contact point and a virtual end-effector to show the direction, solely to better demonstrate RoboMamba's predicted 6-DOF pose, as the physical robotic gripper is quite large and occludes the object. Meanwhile, all the reasoning quantitative results in Figures 4 and 5 are based on real-world data from the RoboVQA validation set. Finally, based on your suggestion, we revised the visualization and provided reasoning failure cases in the **global rebuttal PDF**.
>
> - **_(Weakness 4) Better representation._** Please refer to section **2) Robotic-specific training strategy of Weakness 1**.
>
> - **_(Weakness 5) Ablation study in Table 4._** The LLaVA-v1.5 655K mix dataset is already a carefully selected MLLM training dataset [56]. The introduction of additional datasets beyond LLaVA-v1.5 is specifically aimed at enhancing the model's capabilities. We introduce LRV-INSTRUCT 400K to further prevent hallucinations in robotic tasks, which led to improvements in POPE accuracy. Additionally, we included RoboVQA to empower RoboMamba with robot-related reasoning abilities. Finally, as shown in **global rebuttal Q2 3)**, we explored the impact of different training datasets on manipulation abilities.
>
> - **_(Weakness 6) Scalability._**  We further explore the scalability of RoboMamba by increasing model parameters and training datasets. The results are shown in Table 1 of the **global rebuttal PDF**.
>
> - **_(Questions 1&2)._**
> For manipulation experiments, differences in model architecture and training strategies led to varied performance. For reasoning experiments, different MLLM methods [48,52,56,72] are trained on different datasets, as the design of training strategies are core aspects in the MLLM field. For instance, LLaMA-AdapterV2 [48] trains on COCO Caption and ScienceQA; SPHINX [52] uses additional OCR datasets; TinyLLaVA [72] employs a 1246k pre-training dataset from ShareGPT4V. However, these papers directly compare reasoning capabilities across common benchmarks. We will integrate all your valuable suggestions in the final version.

---

> > ### Comment · Reviewer_16K4 · 2024-08-10
> >
> > Thank you to the author for their additional experiments and clarifications in the rebuttal.
> >
> > First, in my opinion, the engineering effort in this work is good and comprehensive as SSM-based Transformers for robot MMLM is a reasonable thing to try. I would still argue that there is a lack of technical novelty as this seems almost a direct application of a new architecture design to a new problem domain. The benefits of efficiency, linear complexity, etc as described are due to the Mamba architecture. I also want to push back on the claim that combining general VQA data and robot specific-data is "innovative". Again, I feel that it is almost obvious that if we aggregate both types of data, then naturally you'll get the benefits of general reasoning and robot domain knowledge.
> >
> > However, I appreciate the author's efforts in addressing most of my other concerns and the new experiments demonstrating the scalability of their method and I will increase their score.

---

> > > ### Author Response · Authors · 2024-08-10
> > >
> > > Thank you for acknowledging the content and effort of our rebuttal. We also greatly appreciate you upgrading the rating of our paper. In the revised version, we will improve our paper according to your valuable feedback, particularly by including the scalability experiment (Rebuttal PDF) in the main text.  Finally, we will open-source our proposed all-round robotic MLLM system, providing the robotics community with a potential solution that balances high-level reasoning, low-level actions, and efficiency.

---

### Official Review · Reviewer_vV5n · 2024-07-11

**Soundness:** 3
**Presentation:** 3
**Contribution:** 3
**Rating:** 7
**Confidence:** 5

**Summary:**

This paper proposes a Mamba-based framework called RoboMamba that utilizes multimodal state space model for robotic reasoning and manipulation. It addresses the limitations of existing MLLMs in complex reasoning and high computational costs. RoboMamba integrates a vision encoder with the Mamba model to align visual data with language embedding, enabling robust visual reasoning and action capabilities. The model uses a simple lightweight policy head for action pose prediction, requiring minimal fine-tuning. Experiments show RoboMamba excels in reasoning and pose prediction tasks, achieving faster inference speeds and state-of-the-art performance in both simulated and real-world environments.

**Strengths:**

1. Using architecture with linear computational complexity like Mamba for improving efficiency fits quite well with the robotic reasoning and manipulation.
2. The design of training strategy is simple and high-efficient, verifying the transferability of the proposed framework.
3. Experimental results on public benchmarks as well as simulated and real-world environments demonstrate the effectiveness and efficiency of the proposed method.

**Weaknesses:**

1. Besides Mamba-like architecture, there exists many network architectures with linear computational complexity, like XCiT[1], xLSTM[2], hence my first concern is can these architectures work for the proposed framework?
[1] Ali, Alaaeldin, et al. "Xcit: Cross-covariance image transformers." Advances in neural information processing systems 34 (2021): 20014-20027.
[2] Beck, Maximilian, et al. "xLSTM: Extended Long Short-Term Memory." arXiv preprint arXiv:2405.04517 (2024).

2. Following the first question, does Mamba has some unique attributes and advantages specially fit the robotic field? This work lack of related comparison and analysis.

3. The Fig.1 seems a bit of rough, for example, the formats and colors of arrows are inconsistent, some borders have jagged edges, and the overall resolution is not high enough.

**Questions:**

Please refer to the weaknesses of the paper.

**Limitations:**

Please refer to the weaknesses of the paper.

---

> ### Author Rebuttal · Authors · 2024-08-07
>
> - **_Weakness 1. Can other architectures work for the proposed framework._**
>
> Thank you for the constructive comments. We explore whether other architectures could be applied to the proposed framework from two perspectives: 1) replacing the Image Encoder and 2) replacing the LLM.
>
> 1）As you mentioned, we attempt to replace the CLIP encoder used in our submission with a linear computational complexity encoder (XCiT [1]). Meanwhile, we also supplemented the experiment by using SigLIP [c] as an image encoder. As shown in Table 1 below, we explore the impact of different image encoders on reasoning and manipulation abilities. The training dataset and strategy remained consistent across experiments. The results indicate that the choice of image encoder does not significantly affect reasoning ability; rather, the input image resolution is more critical, with higher resolutions improving accuracy. However, using an image encoder without cross-modality alignment (i.e., XCiT) makes it difficult to convert image tokens to LLM language embeddings. Even though our training process includes an alignment pre-training stage, this primarily serves to train the projection layer. Therefore, in future works, we aim to develop a robotic-specific image encoder capable of projecting image tokens into language embeddings while maintaining linear computational complexity. On the other hand, the manipulation results indicate that stronger reasoning abilities in RoboMamba and higher input image resolution aid in policy learning, thereby improving the manipulation accuracy. However, increasing the input resolution also results in additional inference time. Therefore, in our RoboMamba framework, we choose the CLIP (336 x 336) image encoder, which achieves a balance between extracting image semantic knowledge and efficiency.
>
> **Table 1. Ablation Study of the Impact of Different Image Encoders**
> |  | Image Resolution | OKVQA | GQA | POPE | RoboVQA- BLEU-4 | ManipACC(seen) |
> | --- | --- | --- | --- | --- | --- | --- |
> | CLIP | 224 | 63.1 | 62.4 | 85.3 | 36.3 | 52.5 |
> | XCiT [1] | 224 | 46.7 | 50.7 | 79.7 | 35.2 | 41.2 |
> | CLIP | 336 | 62.3 | 63.8 | 86.9 | 40.9 | 63.7 |
> | SigLIP [c] | 384 | 62.4 | 64.4 | 86.0 | 40.6 | 63.4 |
>
> 2）We also attempt to replace Mamba with a linear computational complexity LLM in our proposed framework. However, we could not find official published pre-trained parameters for xLSTM [2] in either 2.7B or 1.3B versions. Therefore, we select RWKV-2.7B [d] and Mamba-1.4B, both with linear complexity. As shown in Table 2 below, we explore the impact of different LLMs on reasoning and manipulation abilities. For all experiments, we use the same training data and strategy. The results show that the Mamba-2.7B model possesses superior visual scene reasoning abilities and can more efficiently learn robotic manipulation skills. For other models, due to the sequence modeling capabilities of the LLM itself, the reasoning and manipulation performance achieved is limited. Therefore, we choose Mamba-2.7B as the LLM for our RoboMamba framework to balance high-level reasoning, low-level manipulation actions, and efficiency. Lastly, we will include all the insightful experiments you suggested in our final version. We will also open-source our codebase, continuously experimenting with the latest linear computational complexity architectures to provide a more efficient MLLM framework for the robotics field.
>
> **Table 2. Ablation Study of the Impact of Different LLMs**
>
> ||Imageresolution|OKVQA|GQA|POPE|RoboVQA-BLEU-4|ManipACC(seen)
> -|-|-|-|-|-|-
> RWKV-3B[d]|224|32.9|41.0|67.1|9.1|46.5
> Mamba-1.4B|224|28.5|40.8|66.8|34.5|50.1
> Mamba-2.7B|224|63.1|62.4|85.3|36.3|52.5
> Mamba-2.7B|336|62.3|63.8|86.9|40.9|63.7
>
>
> - **_Weakness 2. Does Mamba has some unique attributes and advantages specially fit the robotic field?_**
>
> Thank you for the valuable feedback. We would like to explain the robotic-related attributes and advantages of Mamba from the following two perspectives: 1) Balance of Reasoning Ability and Efficiency, and 2) Sequence Modeling Ability.
>
> 1）The design goal of the RoboMamba framework is to simultaneously equip the model with high-level reasoning, low-level manipulation actions, and efficiency, all of which are critical attributes in the robotics field. For reasoning abilities, as shown in Table 2 of the Weakness 1 response shows that RoboMamba exhibits stronger reasoning and manipulation capabilities than other linear LLMs (e.g., RWKV, Phi2). For inference efficiency, which is crucial in robotic manipulation and a major challenge for existing MLLM-based policy methods, RoboMamba-2.7B achieves a control frequency of 9.0 Hz. This is significantly faster than other robotic MLLMs, thanks to Mamba's linear computational complexity.
> Therefore, to balance the attributes needed in the robotics field, we choose Mamba as our LLM, as it possesses both context-aware reasoning ability and efficiency.
>
> 2）Meanwhile, the Mamba architecture demonstrates strong sequence modeling ability [22] by generating hidden-space output tokens autoregressively, which effectively captures complex semantic information. During the robot manipulation fine-tuning phase, we only need to fine-tune a simple and efficient MLP policy head to convert Mamba’s output tokens into action poses, rather than using a complex policy head for decoding. This significantly enhances RoboMamba's learning efficiency when facing new manipulation categories or tasks.
>
> - **_Weakness 3. Figure 1 improvement._**
>
> As shown in the **global rebuttal PDF**, we refined Figure 1 by using arrows with consistent formats and colors, aligning the bounding blocks, and replacing them with a higher-resolution image.
>
> Reference:
> [c][d] (Global rebuttal);

---

> > ### Author Response · Authors · 2024-08-12
> >
> > Dear Reviewer vV5n,
> >
> > Sorry to bother you. We’re sending this comment to confirm whether our response has addressed your concerns. Please feel free to ask any remaining questions, and we’ll respond promptly. Thank you again for your valuable time and comments.
> >
> > Best, Paper 5916 authors

---

> > > ### Comment · Reviewer_vV5n · 2024-08-14
> > > **Post-rebuttal comments**
> > >
> > > Thanks for the effort of making the response. The rebuttal and the supplemented experiments have basically addressed my concerns. The extended ablation studies in response to comments from other reviewers further verify the effectiveness of the proposed methods. Therefore, I will keep my score.

---

> > > > ### Author Response · Authors · 2024-08-14
> > > >
> > > > We are pleased to have addressed your concerns. Based on your valuable feedback, we will incorporate the additional experiments and all extended ablation studies into the final version of our paper. Thank you once again for recognizing our work.

---

### Official Review · Reviewer_iHpb · 2024-07-12

**Soundness:** 3
**Presentation:** 3
**Contribution:** 2
**Rating:** 5
**Confidence:** 3

**Summary:**

This work introduces RoboMamba to leverage SSM model’s capabilities in non-trivial sequence modeling with linear inference complexity. A simple policy head is employed for finetuning to enable RoboMamba to predict action poses. Evaluation is conducted both in simulation(SAPIEN) and real-world settings and shows consistent improvement compared to previous sota methods such as ManipLLM. Besides, fine-tuning parameters and time are significantly reduced and inference speeds achieve 7 times faster than existing models.

**Strengths:**

1. On both general and robotic evaluation benchmarks, RoboMamba on average all shows strong performance.
2. Compared to previous methods, RoboMamber has fewer fine-tuning parameters and time, faster inference speed.

**Weaknesses:**

1. In Table 1, RoboMamba(2.7B)’s performances on MME, MMB, MM-Vet are still below TinyLLaVA’s results.
2. In Table 2, RoboMamba does not achieve sota performance on some of the both training and test categories.
3. Some minor mistakes like: In Figure 3(b), “Openfalmingo”

**Questions:**

1. Can RoboMamba support more complex robot manipulation tasks?
2. For delicate manipulation objects, RoboMamba seems have inferior performance in Table 2. Could you provide some analysis for this?

**Limitations:**

1. As mentioned in Question 1, RoboMamba may not support more complex manipulation tasks.
2. RoboMamba only output 3D poses instead of direct policies.

---

> ### Author Rebuttal · Authors · 2024-08-07
>
> - **_(Weakness 1). Comparison of RoboMamba-2.7B with TinyLLaVA_**
>
> Thank you for the constructive comments. The design goal of RoboMamba is to introduce a new paradigm for adapting Mamba to multimodal robotic tasks, resulting in an innovative Robotic MLLM that integrates high-level reasoning, low-level manipulation actions, and efficiency. Our primary focus is on the robotic domain rather than general multimodal scenarios. We select various general scene MLLM benchmarks to showcase our model's generalizability. However, since RoboMamba and TinyLLaVA use different training datasets, this directly affects their scores on various benchmarks. For example, as shown in Table A1 of TinyLLaVA [81], using ShareGPT4V training datasets with different model architectures can improve MM-Vet performance. Additionally, to more comprehensively compare RoboMamba with TinyLLaVA in the robotic domain, we conducted additional experiments using high-level robotic reasoning benchmarks (RoboVQA) and low-level manipulation pose prediction.
>
> 1）For a fair comparison on RoboVQA, we load TinyLLaVA's parameters and fine-tune it on the RoboVQA training set. As shown in Table 1 below, RoboMamba achieves superior performance across BLEU-1 and BLEU-4, indicating that our model possesses advanced robot-related reasoning capabilities. Meanwhile, TinyLLaVA shows stronger reasoning abilities than LLaMA-AdapterV2 (7B), highlighting its significant potential for robotic-related tasks.
>
> |**Table 1:**|BLEU-1|BLEU-4|
> -|-|-
> LLaMA-AdapterV2(7B)|27.8|8.1
> TinyLLaVA(2.7B)|43.5|29.6
> RoboMamba(2.7B)|54.9|36.3
>
> 2）To compare TinyLLaVA in the manipulation, we use the same fine-tuning strategy by only fine-tuning a simple MLP policy head after the MLLM. As shown in Table 2 below, RoboMamba achieves promising results compared to TinyLLaVA and other methods. These results further support our finding that RoboMamba’s strong reasoning capabilities and framework design enhance the learning of manipulation pose prediction.
>
> |**Table 2:**|LLaMA-AdapterV2|TinyLLaVA|Ours-2.7B
> -|-|-|-
> Acc(Seen)|0.46|0.52|0.63
> Acc(Unseen)|0.20|0.34|0.53
>
> On the other hand, inference efficiency is a crucial attribute for robotic manipulation models and poses a major challenge for existing MLLM-based methods. As shown in **global rebuttal Q1**, RoboMamba-2.7B achieves a control frequency of 9.0 Hz, while TinyLLaVA, under the same inference settings, only achieves 3.9 Hz. These results demonstrate that our approach not only delivers robust performance in the robotic domain but also offers superior efficiency and practicality. Finally, we will incorporate TinyLLaVA's dataset into our co-training stage and update all the aforementioned experiments in the final version.
>
> - **_(Weakness 2 & Question 2). The analysis of Manipulation results_**
>
> We provide a comprehensive analysis of our method's manipulation results compared to the previous SOTA method (ManipLLM), especially for delicate objects. We conduct our analysis from two aspects: 1) different test strategies, and 2) more manipulation fine-tuning parameters.
>
> 1）During inference, after contacting the object with the predicted 6-DOF pose, following where2act [45], our method simply pulls the objects along the predicted z-axis of the end effector pose. In contrast, ManipLLM adopts an 'Active Impedance Adaptation Policy (AIAP)', which heuristically selects the best pulling direction from all proposed directions (i.e., x-axis, y-axis,and z-axis). This strategy significantly increases the testing time, as it requires trying each predicted direction and then selecting the optimal pulling direction. Therefore, we re-evaluated ManipLLM by testing its accuracy in pulling objects along the z-axis. This adjustment resulted in decreased performance, particularly for delicate categories, as shown in Table 3 below. Consequently, we conclude that the improved performance of ManipLLM in some categories is attributed to the heuristic design of the AIAP interaction rather than to the model's learning capabilities.
>
> |**Table 3:**|Display|Mouse|Pliers|Remote|Foldingchair|Toaster|USB|Washing|
> -|-|-|-|-|-|-|-|-
> ManipLLM(z-axis)|0.33|0.35|0.24|0.43|0.36|0.42|0.40|0.56
> RoboMamba(z-axis)|0.33|0.42|0.26|0.39|0.40|0.55|0.45|0.68
>
>
> 2）However, for some delicate categories, RoboMamba still fails to achieve SOTA performance, such as Display, Mouse, Remote, Toilet, and Faucet. This leads us to propose that delicate categories are more challenging to manipulate, so we need to update more parameters during the manipulation learning process. As shown in Table 4 below, increasing the number of MLP layers to 5 improves performance to SOTA levels. Your question is very insightful, and we will continue to explore more detailed experiments.
>
> |**Table4:**|PolicyHead|Display|Mouse|Remote|Toilet|Faucet
> -|-|-|-|-|-|-
> RoboMamba|MLP×2(3.7M)|0.33|0.42|0.39|0.19|0.30
> RoboMamba|MLP×5(9.1M)|0.50|0.50|0.65|0.36|0.42
>
>
> - **_(Question 1 and Limitation 1&2). More complex robot manipulation tasks_**
>
> RoboMamba can support more complex robot manipulation tasks, such as multi-step closed-loop tasks. For the closed-loop benchmark, we selected Meta-World [e], a multi-step tabletop environment. We choose six tasks from Meta-World: Assembly, Bin-Picking, Box Close, Coffee Pull, Hammer, and Button Press. Specifically, we continuously input the current state image to predict the next end-effector pose. Additionally, we utilize an extra MLP to encode the robot's current state and concatenate it with the image features. As shown in **Table 1 of Reviewer CsAK**, RoboMamba still achieves satisfactory results in the relatively more complex multi-step manipulation experiment. For Limitation 2, instead of heuristically setting the manipulation direction, RoboMamba can also control the direction through closed-loop trajectory prediction. Finally, we will present more closed-loop experiments in the final version.
>
> - **_(Weakness 3)_** We will correct all the typos in the final version.

---

> > ### Author Response · Authors · 2024-08-11
> >
> > Dear Reviewer iHpb,
> >
> >
> > As the discussion phase is nearing its conclusion, we would like to confirm whether our response adequately addresses your concerns. Feel free to inquire about any remaining questions, and we'll provide prompt responses. Thank you once again for your valuable comments.
> >
> > Paper 5916 authors

---

> > ### Comment · Reviewer_iHpb · 2024-08-12
> >
> > Thank you for your response to my concerns and the additional experiments you provided which greatly addressed my concerns. I will keep my score. Hopefully, closed-loop experiments will be presented in your final version as you mentioned in your answer to my limitation 2.

---

> ### Author Response · Authors · 2024-08-12
>
> Dear Reviewer iHpb,
>
> We sincerely appreciate your recognition of our work. We will include the closed-loop experiments in the main text and appendix of our paper. May I know if it is possible for you to consider raising your rating above a borderline score if your concerns have been addressed. Thank you very much!
>
> Paper 5916 authors

---

### Official Review · Reviewer_CsAK · 2024-07-13

**Soundness:** 3
**Presentation:** 2
**Contribution:** 3
**Rating:** 6
**Confidence:** 4

**Summary:**

This paper proposes RoboMamba, which applies the Mamba state space model architecture for robotic manipulation policy learning. Prior MLLM-based robot policy learning finetunes transformer-based models and suffers from two major problems: reasoning capabilities degrade with visual input; and training computational cost is high for end-to-end policy learning. RoboMamaba leverages the efficient Mamba architecture to improve learning efficiency and demonstrates simple vision encoder enables visual understanding. Similarly, a simple policy head is shown to be sufficient for generating performant robot behavior after RoboMamba had converged on robotic reasoning tasks.

**Strengths:**

This paper’s major contribution is tailoring a state space model for robotic manipulation control. RoboMamba starts with general pretrained vision and language models and fine-tunes different parts of the network for policy learning while maintaining reasoning capabilities of the model. First, it introduces a visual input encoder for Mamba architecture and demonstrates its effectiveness to enable visual understanding within a pretrained Mamba model. Second, to pretrain the model for policy learning, it trains the model on both general VQA and robot VQA datasets. Last, it learns a light-weight policy head for predicting end-effector poses. Experiments show RoboMamba retains performant reasoning capabilities in VQA tasks while being able to generate competent policies.

**Weaknesses:**

The evaluation of robotic manipulation tasks is limited to one benchmark and contains only single-step open-loop action prediction (instead of learning a closed loop reactive policy). It is unclear how much training data is being used and how much variation there is in testing the policy.  At the same time, this model has many moving parts and design choices that were not carefully ablated.

The presentation of this work can be improved in general for clarity. The authors could better motivate certain design choices as well as evaluation metrics.

**Questions:**

What limits evaluating RoboMamba to predict one-step action? How does it generalize to multi-step or closed loop policy? Is inference spped, i.e. control frequency, the major bottleneck?

How does RoboMamba compare with more structured LLM policies such as Code as Policies [1] and VoxPoser [2]?

[1] https://code-as-policies.github.io/
[2] https://voxposer.github.io/

**Limitations:**

see weakness

---

> ### Author Rebuttal · Authors · 2024-08-07
>
> - **_(Weakness 1 & Question 1). Additional close-loop experiments_**
>
> 1）Thank you for the constructive comments. RoboMamba is not limited to performing single-step open-loop action predictions. In the submitted paper, for fair comparison, we follow the experimental settings of the latest Robotic MLLM (ManipLLM [12]) to implement action pose prediction for articulated objects. RoboMamba can also efficiently generalize to closed-loop policy learning. Specifically, instead of directly predicting the final contact pose of the end-effector, we continuously input the current state image to predict the next end-effector pose. Additionally, we utilize an extra MLP to encode the robot's current state and concatenate it with the image features. For the close-loop benchmark, we select Meta-World [e] to validate RoboMamba's closed-loop action prediction capabilities. Meta-World is a collection of tasks in which agents command a Sawyer robot arm to manipulate objects in a tabletop environment. We consider six tasks from Meta-World: Assembly, Bin-Picking, Box Close, Coffee Pull, Hammer, and Button Press. As shown in Table 1 below, RoboMamba still achieves satisfactory results in the closed-loop experiment. Finally, we will present more closed-loop experiments in the final version.
>
> _**Table 1. The closed-loop experiments of RoboMamba, with the success rate metric.**_
>
> ||Assembly|Bin-Picking|Boxclose|CoffeePull|Hammer|ButtonPress
> -|-|-|-|-|-|-
> MVP[f]|0.89|0.79|0.58|0.62|0.98|0.70
> CLIP[23]|0.71|0.68|0.72|0.80|0.90|0.48
> RoboMamba|0.98|0.80|0.80|0.94|1.0|0.98
>
> 2）Since the Mamba LLM used in RoboMamba balances context-aware reasoning ability with linear computational complexity, inference speed is not a limiting factor for RoboMamba in learning closed-loop policies. We provide detailed inference speed information in **global rebuttal Q1**.
>
> - **_(Weakness 1). How much training data is being used and how much variation there is in testing the policy._**
>
> The size of the training dataset (10K) is described in Lines 218-223 of the submitted paper. We provide details of the simulator data collection and categories in Lines 603-612 of Appendix B. Regarding the variation between training and testing data, we followed the data collection settings of where2act [45] and ManipLLM [12]. The specific variations can be divided into two aspects: 1) asset variation and 2) state variation.
>
> **1) Asset Variation**: We use 20 categories from PartNet [42] for seen objects and reserve the remaining 10 categories for unseen objects to analyze if RoboMamba can generalize to noval categories. Specifically, we further divide the seen objects into 1037 training shapes and 489 testing shapes, using only the training shapes to construct the training data. Thus, the shapes of the seen objects encountered during training and testing are different. For unseen categories, there are a total of 274 shapes, which are used exclusively in the testing data.
>
> **2) State Variation**: We observe the object in the scene from an RGB-D camera with known intrinsics, mounted 4.5-5.5 units away from the object, facing its center. The camera is located at the upper hemisphere of the object with a random azimuth between [-45, 45] and a random altitude between [30, 60]. Since the tasks involve 'pulling,' we also initialize the starting pose for each articulated part randomly between its rest joint state (fully closed) and any position up to half of its joint state (half-opened). These state settings are utilized for both training and testing data, aiming to boost the model's generalization ability.
>
> - **_(Weakness 1 & 2). Additional Ablation Study of Moving Parts and Design Choices_**
>
> The submitted paper includes ablation studies such as the impact of MLLM reasoning ability on manipulation policy learning (Figure 3b), the effect of using different training datasets on reasoning ability (Table 4), and the impact of policy head parameters on manipulation accuracy (Table 5). However, based on your valuable comments, we find that some ablation studies are missing. To address this, we explore the impact of different image encoders and LLMs on the reasoning and manipulation capabilities of our proposed framework, while also examining the effect of using different training datasets on manipulation accuracy. Due to space limitations, please refer to **global rebuttal Q2**, which includes 1) Image Encoder Ablation, 2) Large Language Models (LLMs) Ablation, and 3) The Impact of Training Strategies on Manipulation Abilities.
>
> For evaluation metrics, we select several MLLM benchmarks to comprehensively assess our model’s reasoning capabilities and generalizability, prioritizing those related to robotics. Detailed descriptions can be found in Lines 647-665 of Appendix E. For manipulation, we follow previous works [41, 51, 12] and selected manipulation successful accuracy as the evaluation metric, as described in Lines 240-246 of Section 4.1. Finally, we will include all the aforementioned ablation studies in the final version and conduct more analysis for each moving part.
>
> - _**(Question 2). Comparisons with structured LLM policy method.**_
>
> Due to time limitations, we reproduce the more recent VoxPoser method in the SAPIEN environment to evaluate its performance on tasks identical to ours, including 'Open Drawer,' 'Open Door,' and 'Open Faucet.' The manipulation success rates achieved by VoxPoser are 0.19, 0.36, and 0.12, respectively, which are lower compared to our results (i.e., 0.86, 0.73, and 0.30). Notably, while VoxPoser employs motion planning based on affordance and obstacle maps generated by an LLM, it still lacks comprehensive visual scene understanding. In contrast, RoboMamba possesses strong visual scene and robotic reasoning capabilities, conditioned on both image and text inputs. Therefore, RoboMamba demonstrates more robust performance in our experiments.
>
> Reference:
> [e](Global rebuttal)
> [f] Real world robot learning with masked visual pre-training

---

> > ### Author Response · Authors · 2024-08-11
> >
> > Dear reviewer CsAK,
> >
> > As the discussion phase is nearing its close, we would like to confirm whether our response adequately addresses your concerns. If you have any remaining questions, please don't hesitate to ask, and we'll respond promptly. Thank you for your valuable time and insightful comments.
> >
> > Paper 5916 authors

---

> ### Comment · Reviewer_CsAK · 2024-08-12
> **thanks for the additional experiments**
>
> Thanks the authors for clarifying my concerns and adding the simulation experiments for closed-loop policy learning and comparing with VoxPoser. I am happy to raise my score to 6 but suggest the authors add additional comprehensive real-world experiments to have a broader impact.

---

> > ### Author Response · Authors · 2024-08-12
> >
> > Thank you for acknowledging our work and rebuttal. We greatly appreciate your decision to upgrade the rating of our paper. We will incorporate all of your valuable suggestions into the revised version. Additionally, we will include the quantitative results of real-world experiments in the main text or appendix of our paper.

---

### Official Review · Reviewer_QK1g · 2024-07-14

**Soundness:** 3
**Presentation:** 3
**Contribution:** 3
**Rating:** 6
**Confidence:** 4

**Summary:**

The paper proposes a method for robot reasoning and manipulation by developing a multimodal large language model (MLLM) based on the Mamba state space model (SSM). The joint model is able to both attack robot reasoning tasks assessed via VQA, and solve robot manipulation tasks by fine-tuning a dedicated head that contains only a small fraction of the model parameters.

**Strengths:**

The paper introduces an MLLM based on the Mamba model built specifically for robotic tasks. The exposition of the method is clear and well motivated. The proposed method takes advantage of the linear complexity of Mamba and, in combination to the reduced number of parameters used for the manipulation policy head, achieves higher efficiency and better performance with respect to state-of-the-art robot MLLM methods. Another strong point is the combination of natural language reasoning abilities and task specific manipulation policies based on the MLLM output.

The experimental evaluation is comprehensive, considering both simulated and real-world experiments, where the proposed method achieves competitive results with respect to SOTA methods across multiple benchmarks.

**Weaknesses:**

Although time efficiency is one of the major claimed contributions of the work, the aspect of time efficiency is not well covered in the experimental evaluation. The fact that the proposed method is 7 times faster than SOTA MLLM methods is mentioned a couple of times, however detailed results regarding inference times are not provided.

Another aspect not covered in detail, regards the effect the number of parameters in the manipulation head has on the performance and efficiency of the manipulation task.

Another question that comes to mind, is whether it is possible to apply the idea of fine-tuning only a small-sized manipulation head to other MLLMs. Does Mamba work better with this constrained policy head and, if yes, why?

Finally, another issue regards reproducibility, as details regarding the structure of the projection layers and the MLPs used in the manipulation policy head are not provided.

### Minor comments
- L.13: "action pose prediction abilities" is not very clear
- L.100 "ManipVQA, enhancing robotic" verb missing
- L.192: what is the angle representation considered?
- L.235: "on when"

**Questions:**

- Can you please provide results regarding comparison with SOTA methods in terms of time efficiency?
- Can the training strategy, including the fine-tuning of only the manipulation policy head, be applied to other MLLMs?
- Is the performance in the manipulation tasks affected by the number of parameters in the corresponding head and how?

**Limitations:**

Limitations are discussed in the text.

---

> ### Author Rebuttal · Authors · 2024-08-07
>
> - **_(Weakness 1 & Question 1). Detailed inference speed comparison_**
>
> Thank you for the constructive comments. Inference efficiency is a crucial evaluation metric in robotic manipulation and poses a major challenge for existing Multimodal Large Language Model (MLLM) based policy models. We compare the control frequency (Hz) of our proposed RoboMamba with previous robotic MLLM. All inferences are conducted on the NVIDIA A100 GPU without any quantization or inference speed-up techniques.
>
> **Control frequency (Hz).** This measures the number of times per second the model can complete inference and generate a manipulation pose. As shown in Table 1 below, RoboMamba-2.7B achieves a control frequency of 9.0 Hz even with the highest input image resolution (336 x 336). These results demonstrate that RoboMamba not only possesses robust reasoning and manipulation capabilities but also maintains efficient inference speed.
>
> Table 1: Comparison of control frequency with previous robotic MLLMs
> |  | Input Resolution | Parameters | Control Frequence（Hz） |
> | --- | --- | --- | --- |
> | RT-1 | 300 x 300 | 35M | 3 Hz |
> | Open-VLA [a] | 224 x 224  | 7B | 3.4 Hz |
> | ManipLLM | 336 x 336 | 7B | 0.7 Hz |
> | RoboMamba | 336 x 336 | 2.7B | 9.0 Hz |
>
> Meanwhile, we compared the token output speed (tokens/s) with ManipLLM. This indicates the efficiency of Robotic MLLM in performing language output tasks (e.g., task planning or visual question answering). Specifically, we compare the number of tokens output per second by each model for the same question. ManipLLM can generate 133.1 language tokens per second after receiving 336 x 336 input images and a question. RoboMamba-2.7B is more efficient, generating 898.4 language tokens per second. These results demonstrate that our model maintains efficient inference speed when answering robotic high-level reasoning tasks using language responses.
>
> - **_(Weakness 2 & Question 3). The effect of manipulation head parameters on performance and efficiency_**
>
> We have investigated the impact of manipulation head parameters on performance by conducting experiments detailed in Lines 625-634 and Table 5 of Appendix C. The findings reveal that the manipulation success rate across the three configurations is similar. This suggests that RoboMamba, due to its sufficient robotic reasoning capabilities, can achieve effective pose prediction skills at a low cost, as the impact of policy head parameters is not sensitive. Meanwhile, as shown in Table 2 below, the effect of manipulation head parameters on efficiency is negligible, since the number of head parameters is very small compared to the overall RoboMamba parameters.
>
> Table 2. The effect of the number of parameters in the manipulation head
> |  | MLPx2 | MLPx1 | (SSM block+MLP)x2 |
> | - | - | - | - |
> | Accuracy(Seen) | 63.7 | 62.1 | 63.2
> | Parameters | 3.7M | 1.8M | 45.2M
> | Percentage | 0.11% | 0.05% | 1.3%
> | Frequency（Hz） | 9.0 Hz | 9.0 Hz |8.6 Hz
>
> - **_(Weakness 3 & Question 2). Fine-tuning only a small-sized manipulation head on other MLLMs?_**
>
> In Lines 296-308 and Figure 3 b) of the submission, we have conducted experiments applying the idea of fine-tuning only a small-sized manipulation head to other MLLMs. For convenience, we present the results in Table 3 below, including additional experiments with TinyLLaVA [72]. With the same fine-tuned policy head and training dataset, our RoboMamba-2.7B achieves promising results compared to other MLLMs. These results demonstrate our finding: if MLLMs possess strong robot-related reasoning abilities, they can be efficiently fine-tuned to learn robot manipulation skills. Meanwhile, the Mamba architecture exhibits strong sequence modeling ability [22]; it produces hidden-space output tokens in an autoregressive manner, effectively capturing complex semantic and spatial information in visual scenes. Therefore, RoboMamba can only fine-tune a simple and efficient MLP policy head to convert Mamba’s output tokens into action poses, rather than using a complex policy head for decoding. This significantly enhances RoboMamba's learning efficiency when facing new manipulation categories or tasks.
>
> Table 3. Fine-tuning only a small-sized manipulation head to other MLLMs
> | | Openflamingo | LLaMA-AdapterV2 | TinyLLaVA | Ours-1.4B | Ours-2.7B
>  -|-| -|-|-|-
> Acc(Seen) | 0.26 | 0.46 | 0.51 | 0.39 | 0.63
> Acc(Unseen) | 0.33 | 0.20 | 0.34 | 0.41 | 0.53
>
>
> - **_(Weakness 4). The structure of projection layers and the MLPs_**
>
> Each MLP used in the projection layer, position head, and direction head consists of two linear layers with a ReLU activation function in between. For the projection layer, the input dimension for the first linear layer is B×N×1024, and the output dimension remains B×N×1024, where B and N represent batch size and tokens, respectively. The input dimension for the second linear layer is B×N×1024, and the output dimension is B×N×2560. For the position head and direction head, both have input dimensions of B×N×128 for the first linear layer, and the output dimensions are B×N×128. The input dimensions for the second linear layer are B×N×128, and the output dimensions are B×N×3, collectively forming the 6-DOF poses of the end-effector. We will publish all the code to ensure RoboMamba's reproducibility.
>
> - **_(Weakness 5). Minor comments_**
>
> Thank you for your detailed comments; we will fix all the typos in the final version. For example: 1) We will revise Line 13 to: "To further enhance RoboMamba with action pose predictions for robot control in SE(3) space, we explore an efficient fine-tuning strategy using a simple policy head." 2) We will revise Line 100 to: "ManipVQA enhances robotic manipulation with physically grounded information processed by MLLM." 3) In Line 192, we utilize rotation matrices to represent pose direction. 4) We will revise Line 235 to: " As detailed in Appendix E, we describe the key aspects each benchmark considers when assessing models in the field of robotics."

---

> > ### Author Response · Authors · 2024-08-12
> >
> > Dear Reviewer QK1g,
> >
> > As the discussion phase progresses, we would like to confirm whether our response has addressed your concerns. If you have any remaining questions, we would be happy to discuss and address them. Thank you once again for your valuable feedback.
> >
> > Best, Paper 5916 authors

---

> ### Comment · Reviewer_QK1g · 2024-08-13
> **Post-rebuttal comments**
>
> I thank the authors for their answers and clarifications. I think the additional experiments and results presented in their answers make the paper stronger, and they should be included in the final version of the paper. I retain my original rating, suggesting acceptance.

---

> ### Author Response · Authors · 2024-08-13
>
> We greatly appreciate your recognition of our work. We will incorporate the additional experiments and detailed module descriptions into the final version. Thank you once again for your valuable time and feedback.

---

### Author Rebuttal · Authors · 2024-08-07

**To all the reviewers:** First, we greatly appreciate all the reviewers' valuable comments and time. Due to character limits in the separate responses, we address some of the reviewers' questions in this global rebuttal. Please review the individual rebuttal response first and then come back to this global rebuttal; we have clearly indicated the points that refer to the global rebuttal. We hope our responses have resolved all your questions. If there are any unclear explanations, please kindly provide further comments and we are committed to promptly addressing them and providing you with a comprehensive response.

_**(To all the reviewers) Q1. Detailed Inference Speed Comparison with Other Robotic MLLMs**_
The design goal of RoboMamba is to introduce a new paradigm for adapting Mamba to multimodal robotic tasks, resulting in an innovative Robotic MLLM that integrates high-level reasoning, low-level manipulation actions, and efficiency. Inference efficiency is a crucial attribute in robotic manipulation and poses a major challenge for existing Multimodal Large Language Model (MLLM) based policy models. Therefore, we compare the control frequency (Hz) of our proposed RoboMamba with previous robotic MLLMs. All inferences are conducted on the NVIDIA A100 GPU without any quantization or inference speed-up techniques. The control frequency measures the number of times per second the model can complete inference and generate a manipulation pose. As shown in Table 1 below, RoboMamba-2.7B achieves a control frequency of 9.0 Hz even with the highest input image resolution. These results demonstrate that RoboMamba not only possesses robust reasoning and manipulation capabilities but also maintains efficient inference speed. The efficient reasoning speed makes RoboMamba more practical and scalable in a wider range of robotic downstream tasks.

_**Table 1: Comparison of control frequency with previous robotic MLLMs**_

|  |Input resolution|Parameters|control frequency(Hz)
-|-|-|-
RT-1[10]|300| 35M|3 Hz
Open VLA[a]|224|7B|3.4 Hz
ManipLLM[12]|336|7B|0.7 Hz
RoboMamba|336|2.7B|9.0 Hz



_**(To reviewer CsAK and 16K4) Q2. More ablation study on moving parts and design choices**_

**1) Image Encoder Ablation**: As shown in Table 2 below, we explore the impact of different image encoders on reasoning and manipulation accuracy. The training dataset and strategy remained consistent across experiments. XCiT [b] is an image encoder with linear computational cost that has not undergone cross-modality alignment similar to CLIP and SigLIP [c]. The results demonstrate that the choice of image encoder does not significantly affect reasoning ability; rather, the input image resolution is more critical, with higher resolutions improving accuracy. However, using an image encoder without cross-modality alignment (i.e., XCiT) makes it difficult for Mamba LLM to comprehend image features, resulting in poorer performance. Meanwhile, the manipulation results indicate that stronger reasoning abilities in RoboMamba and higher input image resolution aid in policy learning, thereby improving the final manipulation accuracy.

_**Table 2. Ablation Study on the Impact of Different Image Encoders**_

| |Image resolution|OKVQA|GQA|POPE|RoboVQA-BLEU-4|Manip ACC(seen)
-|-|-|-|-|-|-
CLIP|224|63.1|62.4|85.3|36.3|52.5
XCiT[b]|224|46.7|50.7|79.7|35.2|41.2
CLIP|336|62.3|63.8|86.9|40.9|63.7
SigLIP [c]|384|62.4|64.4|86.0|40.6|63.4


_**2) Large Language Models Ablation**_: As shown in Table 3 below, we explore the impact of different LLMs on reasoning and manipulation abilities. Given that efficiency is crucial in robotic tasks and directly affects the practicality of policy models, we compare Mamba-2.7B with other linear complexity LLMs. For all experiments, we utilize the same training data and strategy. The results demonstrate that the Mamba-2.7B model not only possesses linear complexity but also efficiently gains visual reasoning abilities through our proposed training strategy. Meanwhile, after robot manipulation fine-tuning, it achieves outstanding manipulation accuracy. Therefore, we chose Mamba-2.7B as the LLM for our RoboMamba framework to simultaneously provide high-level reasoning, low-level manipulation actions, and efficiency.

_**Table 3. The Impact of Different LLMs**_

||Imageresolution|OKVQA|GQA|POPE|RoboVQA-BLEU-4|ManipACC(seen)
-|-|-|-|-|-|-
RWKV-3B[d]|224|32.9|41.0|67.1|9.1|46.5
Mamba-1.4B|224|28.5|40.8|66.8|34.5|50.1
Mamba-2.7B|224|63.1|62.4|85.3|36.3|52.5
Mamba-2.7B|336|62.3|63.8|86.9|40.9|63.7

_**3) The impact of training strategies on manipulation abilities:**_
In Lines 614-624 and Table 4 of the submitted paper, we explored the impact of different training strategies and datasets on RoboMamba’s reasoning abilities. As shown in Table 4 below, we further examine the impact of various training datasets on manipulation abilities. In this table, AP refers to alignment pre-training, and IC refers to instruction co-training with different dataset combinations. The results indicate that each dataset contributed to improved manipulation policy learning, particularly with the application of the RoboVQA dataset. This further supports our finding that when RoboMamba possesses sufficient reasoning capabilities, it can facilitate the learning of low-level manipulation, especially when it has robotic-related reasoning abilities.

_**Table 4. Ablation study of training strategies on robot manipulation benchmark.**_

||AP|IC(LLaVA-655K)|IC(LRV-400k)|IC(Robo-800k)|GQA|POPE|Manipulation ACC(seen)|
-|-|-|-|-|-|-|-
|Ex1||✓|||62.2|85.5|58.2|
|Ex2|✓|✓|||62.7|85.9|58.8|
|Ex3|✓|✓|✓||62.6|86.6|60.3|
|Ex4|✓|✓|✓|✓|63.8|86.9|63.7|

Reference:
[a] OpenVLA: An Open-Source Vision-Language-Action Model;
[b] Xcit: Cross-covariance image transformers;
[c] Sigmoid loss for language image pre-training;
[d] RWKV: Reinventing RNNs for the Transformer Era;
[e] Meta-world: A benchmark and evaluation for multi-task and meta reinforcement learning.

---

### Decision · Program_Chairs · 2024-09-25

**Decision:**

Accept (poster)

**Comment:**

The submission initially received mixed reviews; the authors did a great job during the rebuttal, after which all reviewers became positive about the submission.  The AC agrees with the recommendations.  The authors should incorporate the rebuttal into the camera ready.